# An estimate of the deepest branches of the tree of life from ancient vertically evolving genes

Edmund RR Moody[1], Tara A Mahendrarajah[2], Nina Dombrowski[2], James W Clark[1], Celine Petitjean[1], Pierre Offre[2], Gergely J Szöllősi[3,4,5], Anja Spang[2,6]*, Tom A Williams[1]*

[1]School of Biological Sciences, University of Bristol, Bristol, United Kingdom; [2]NIOZ, Royal Netherlands Institute for Sea Research, Department of Marine Microbiology and Biogeochemistry, Den Burg, Netherlands; [3]Department of Biological Physics, Eötvös Loránd University, Budapest, Hungary; [4]MTA-ELTE "Lendület" Evolutionary Genomics Research Group, Budapest, Hungary; [5]Institute of Evolution, Centre for Ecological Research, Budapest, Hungary; [6]Department of Cell- and Molecular Biology, Science for Life Laboratory, Uppsala University, Uppsala, Sweden

*For correspondence:
Anja.Spang@nioz.nl (AS);
tom.a.williams@bristol.ac.uk
(TAW)

**Abstract** Core gene phylogenies provide a window into early evolution, but different gene sets and analytical methods have yielded substantially different views of the tree of life. Trees inferred from a small set of universal core genes have typically supported a long branch separating the archaeal and bacterial domains. By contrast, recent analyses of a broader set of non-ribosomal genes have suggested that Archaea may be less divergent from Bacteria, and that estimates of inter-domain distance are inflated due to accelerated evolution of ribosomal proteins along the inter-domain branch. Resolving this debate is key to determining the diversity of the archaeal and bacterial domains, the shape of the tree of life, and our understanding of the early course of cellular evolution. Here, we investigate the evolutionary history of the marker genes key to the debate. We show that estimates of a reduced Archaea-Bacteria (AB) branch length result from inter-domain gene transfers and hidden paralogy in the expanded marker gene set. By contrast, analysis of a broad range of manually curated marker gene datasets from an evenly sampled set of 700 Archaea and Bacteria reveals that current methods likely underestimate the AB branch length due to substitutional saturation and poor model fit; that the best-performing phylogenetic markers tend to support longer inter-domain branch lengths; and that the AB branch lengths of ribosomal and non-ribosomal marker genes are statistically indistinguishable. Furthermore, our phylogeny inferred from the 27 highest-ranked marker genes recovers a clade of DPANN at the base of the Archaea and places the bacterial Candidate Phyla Radiation (CPR) within Bacteria as the sister group to the Chloroflexota.

## Editor's evaluation

This contribution is of interest to molecular phylogeny scientists, in particular, and to a broad public interested in early evolution, in general, as it elegantly supports the long-standing (but recently challenged) hypothesis that bacteria and archaea are separated by a long branch.

## Introduction

Much remains unknown about the earliest period of cellular evolution and the deepest divergences in the tree of life. Phylogenies encompassing both Archaea and Bacteria have been inferred from a

'universal core' set of 16–56 genes encoding proteins involved in translation and other aspects of the genetic information processing machinery (*Ciccarelli et al., 2006*; *Fournier and Gogarten, 2010*; *Harris et al., 2003*; *Hug et al., 2016*; *Koonin, 2003*; *Mukherjee et al., 2017*; *Petitjean et al., 2014*; *Ramulu et al., 2014*; *Raymann et al., 2015*; *Theobald, 2010*; *Williams et al., 2020*). While representing a small fraction of the total genome of any organism (*Dagan and Martin, 2006*), these genes are thought to predominantly evolve vertically and are thus best suited for reconstructing the tree of life (*Ciccarelli et al., 2006*; *Creevey et al., 2011*; *Puigbò et al., 2009*; *Ramulu et al., 2014*; *Theobald, 2010*). In these analyses, the branch separating Archaea from Bacteria (hereafter, the AB branch) is often the longest internal branch in the tree (*Cox et al., 2008*; *Gogarten et al., 1989*; *Hug et al., 2016*; *Iwabe et al., 1989*; *Pühler et al., 1989*; *Williams et al., 2020*). In molecular phylogenetics, branch lengths are usually measured in expected numbers of substitutions per site, with a long branch corresponding to a greater degree of genetic change. Long branches can therefore result from high evolutionary rates, long periods of absolute time, or a combination of the two. If a sufficient number of fossils are available for calibration, molecular clock models can, in principle, disentangle the contributions of these effects. However, limited fossil data (*Sugitani et al., 2015*) is currently available to calibrate early divergences in the tree of life (*Betts et al., 2018*; *Horita and Berndt, 1999*; *Lepland et al., 2002*; *van Zuilen et al., 2002*), and as a result, the ages and evolutionary rates of the deepest branches of the tree remain highly uncertain.

Recently, *Zhu et al., 2019* inferred a phylogeny from 381 genes distributed across Archaea and Bacteria using the supertree method ASTRAL (*Mirarab et al., 2014*). These markers increase the total number of genes compared to other universal marker sets and comprise not only proteins involved in information processing but also proteins affiliated with most other functional COG categories, including metabolic processes (*Supplementary file 1*). The genetic distance (AB branch length) between the domains (*Zhu et al., 2019*) was estimated from a concatenation of the same marker genes, resulting in a much shorter AB branch length than observed with the core universal markers (*Hug et al., 2016*; *Williams et al., 2020*). These analyses were consistent with the hypothesis (*Petitjean et al., 2014*; *Zhu et al., 2019*) that the apparent deep divergence of Archaea and Bacteria might be the result of an accelerated evolutionary rate of genes encoding translational and in particular ribosomal proteins along the AB branch as compared to other genes. Interestingly, the same observation was made previously using a smaller set of 38 non-ribosomal marker proteins (*Petitjean et al., 2014*), although the difference in AB branch length between ribosomal and non-ribosomal markers in that analysis was reported to be substantially lower (roughly twofold, compared to roughly 10-fold for the 381 protein set [*Petitjean et al., 2014*; *Zhu et al., 2019*]).

A higher evolutionary rate of ribosomal genes might result from the accumulation of compensatory substitutions at the interaction surfaces among the protein subunits of the ribosome (*Petitjean et al., 2014*; *Valas and Bourne, 2011*) or as a compensatory response to the addition or removal of ribosomal subunits early in evolution (*Petitjean et al., 2014*). Alternatively, differences in the inferred AB branch length might result from varying rates or patterns of evolution between the traditional core genes (*Spang et al., 2015*; *Williams et al., 2020*) and the expanded set (*Zhu et al., 2019*). Substitutional saturation (multiple substitutions at the same site) (*Jeffroy et al., 2006*) and across-site compositional heterogeneity can both impact the inference of tree topologies and branch lengths (*Foster, 2004*; *Lartillot et al., 2007*; *Lartillot and Philippe, 2004*; *Quang et al., 2008*; *Wang et al., 2008*; *Williams et al., 2021*). These difficulties are particularly significant for ancient divergences (*Gouy et al., 2015*). Failure to model site-specific amino acid preferences has previously been shown to lead to underestimation of the AB branch length due to a failure to detect convergent changes (*Tourasse and Gouy, 1999*; *Williams et al., 2020*), although the published analysis of the 381 marker set did not find evidence of a substantial impact of these features on the tree as a whole (*Zhu et al., 2019*). Those analyses also identified phylogenetic incongruence among the 381 markers, but did not determine the underlying cause (*Zhu et al., 2019*).

This recent work (*Zhu et al., 2019*) raises two important issues regarding the inference of the universal tree: first, that estimates of the genetic distance between Archaea and Bacteria from classic 'core genes' may not be representative of ancient genomes as a whole, and second, that there may be many more suitable genes to investigate early evolutionary history than generally recognized, providing an opportunity to improve the precision and accuracy of deep phylogenies. Here, we investigate these issues in order to determine how different methodologies and marker sets affect estimates

of the evolutionary distance between Archaea and Bacteria. First, we examine the evolutionary history of the 381-gene marker set (hereafter, the expanded marker gene set) and identify several features of these genes, including instances of inter-domain gene transfers and mixed paralogy, that may contribute to the inference of a shorter AB branch length in concatenation analyses. Then, we re-evaluate the marker gene sets used in a range of previous analyses to determine how these and other factors, including substitutional saturation and model fit, contribute to inter-domain branch length estimations and the shape of the universal tree. Finally, we identify a subset of marker genes least affected by these issues and use these to estimate an updated tree of the primary domains of life and the length of the branch that separates Archaea and Bacteria.

## Results and discussion

### Genes from the expanded marker set are not widely distributed in Archaea

The 381-gene set was derived from a larger set of 400 genes used to estimate the phylogenetic placement of new lineages as part of the PhyloPhlAn method (*Segata et al., 2013*) and applied to a taxonomic selection that included 669 Archaea and 9906 Bacteria (*Zhu et al., 2019*). Perhaps reflecting the focus on Bacteria in the original application, the phylogenetic distribution of the 381 marker genes in the expanded set varies substantially (*Supplementary file 1*), with many being poorly represented in Archaea. Specifically, 41% of the published gene trees (https://biocore.github.io/wol/; *Zhu et al., 2019*) contain less than 25% of the sampled archaea, with 14 and 68 of these trees including 0 or ≤10 archaeal homologues, respectively. Across all of the gene trees, archaeal homologues comprise 0–14.8% of the dataset (*Supplementary file 1*). Manual inspection of subsampled versions of these gene trees suggested that 317/381 did not possess an unambiguous branch separating the archaeal and bacterial domains (*Supplementary file 1*). These distributions suggest that many of these genes are not broadly present in both domains, and that some might be specific to Bacteria.

### Conflicting evolutionary histories of individual marker genes and the inferred species tree

In the published analysis of the 381-gene set (*Zhu et al., 2019*), the tree topology was inferred using the supertree method ASTRAL (*Mirarab et al., 2014*), with branch lengths inferred on this fixed tree from a marker gene concatenation (*Zhu et al., 2019*). The topology inferred from this expanded marker set (*Zhu et al., 2019*) is similar to previous trees (*Castelle and Banfield, 2018*; *Hug et al., 2016*) and recovers Archaea and Bacteria as reciprocally monophyletic domains, albeit with a shorter AB branch than in earlier analyses. However, the individual gene trees (*Zhu et al., 2019*) differ regarding domain monophyly: Archaea and Bacteria are recovered as reciprocally monophyletic groups in only 22 of the 381 published (*Zhu et al., 2019*) maximum likelihood (ML) gene trees of the expanded marker set (*Supplementary file 1*).

Since single-gene trees often fail to strongly resolve ancient relationships, we used approximately unbiased (AU) tests (*Shimodaira, 2002*) to evaluate whether the failure to recover domain monophyly in the published ML trees is statistically supported. For computational tractability, we performed these analyses on a 1000-species subsample of the full 10,575-species dataset that was compiled in the original study (*Zhu et al., 2019*). For 79 of the 381 genes, we could not perform the test because the gene family did not contain any archaeal homologues (56 genes) or contained only one archaeal homologue (23 genes); in total, the 1000-species sample included 74 archaeal genomes. For the remaining 302 genes, domain monophyly was rejected at the 5% significance level (with Bonferroni correction, $p<0.0001656$) for 151 out of 302 (50%) genes. As a comparison, we performed the same test on several smaller marker sets used previously to infer a tree of life (*Coleman et al., 2021*; *Petitjean et al., 2014*; *Williams et al., 2020*); none of the markers in those sets rejected reciprocal domain monophyly ($p<0.05$ for all genes, with Bonferroni correction: Coleman: $>0.001724$; Petitjean: $>0.001316$; Williams: $>0.00102$: *Figure 1A*). In what follows, we refer to four published marker gene sets as (i) the expanded set (381 genes; *Zhu et al., 2019*); (ii) the core set (49 genes; *Williams et al., 2020*), encoding ribosomal proteins and other conserved information-processing functions; itself a consensus set of several earlier studies (*Da Cunha et al., 2017*; *Spang et al., 2015*; *Williams et al., 2012*); (iii) the non-ribosomal set (38 genes, broadly distributed and explicitly selected to avoid genes

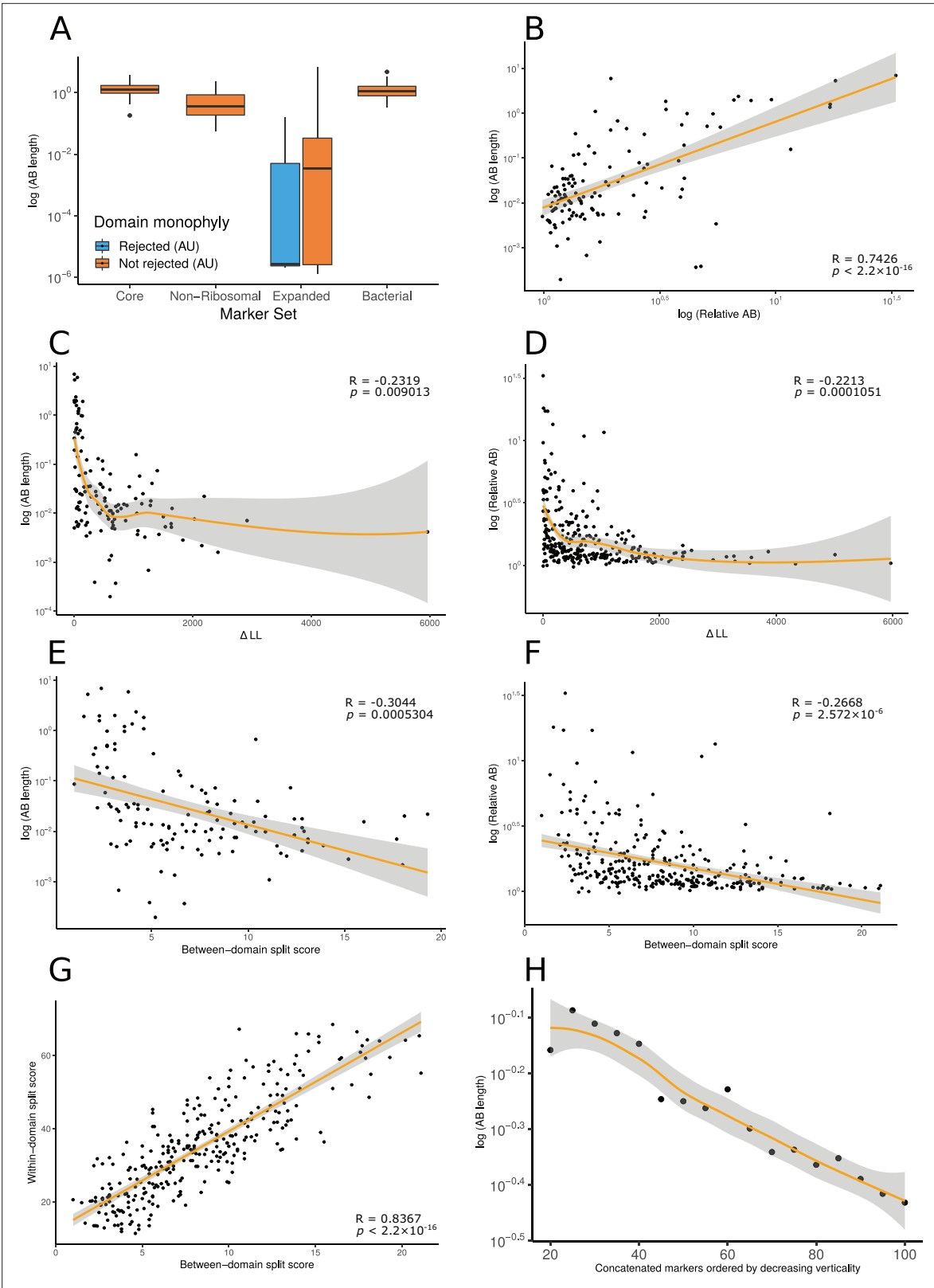

**Figure 1.** Vertically evolving marker genes support a greater evolutionary distance between Archaea and Bacteria. (**A**) Expanded set genes that reject domain monophyly (p<0.05, approximately unbiased [AU] test, with Bonferroni correction [see main text]) support significantly shorter Archaea-Bacteria (AB) branch lengths when constrained to follow a domain monophyletic tree (p=3.653 × 10⁻⁶, Wilcoxon rank-sum test). None of the marker genes from several other published analyses significantly reject domain monophyly (Bonferroni-corrected p<0.05, AU test) for all genes tested, consistent

*Figure 1 continued on next page*

*Figure 1 continued*

with vertical inheritance from the LUCA (last universal common ancestor) to the last common ancestors of Archaea and Bacteria, respectively. (**B**) Two measures of evolutionary proximity (*Zhu et al., 2019*), AB branch length and relative AB distance, are positively correlated ($R = 0.7426499$, $p < 2.2 \times 10^{-16}$). We considered two complementary proxies of marker gene verticality: ΔLL (**C**: against AB branch length; **D**: against relative AB length), which reflects the degree to which marker genes reject domain monophyly (**C**: $p = 0.009013$ and $R = -0.2317894$; **D**: $p = 0.0001051$ and $R = -0.2213292$); and the between-domain split score (**E**: against AB branch length; **F**: against relative AB length), which quantifies the extent to which marker genes recover monophyletic Archaea and Bacteria; a higher split score (see Materials and methods) indicates the splitting of domains into multiple gene tree clades due to gene transfer, reciprocal sorting out of paralogs, or lack of phylogenetic resolution (**E**: $p = 0.0005304$ and $R = -0.3043537$; **F**: $p = 2.572 \times 10^{-6}$ and $R = -0.2667739$). We also considered a split score based on within-domain relationships (**G**); between- and within-domain split scores are positively correlated: $R = 0.836679$, $p < 2.2 \times 10^{-16}$, Pearson's correlation, indicating that markers that recover Archaea and Bacteria as monophyletic also tend to recover established within-domain relationships. (**H**) Inferred AB length decreases as marker genes of lower verticality (larger ΔLL) are added to the concatenate. Marker genes were sorted by ΔLL, the difference in log-likelihood between the maximum likelihood gene family tree under a free topology search and the log-likelihood of the best tree constrained to obey domain monophyly. Note that 79/381 expanded set markers had zero or one archaea in the 1000-species subsample and so could not be included in these analyses; of the remaining 302 markers, 176 have AB branch lengths very close to 0 in the constraint tree (as seen in panel **A**). In these plots, we removed all markers with an AB branch length of <0.00001; see *Figure 1—figure supplements 1–13* for all plots. Nonlinear trendlines were estimated using LOESS regression.

The online version of this article includes the following figure supplement(s) for figure 1:

**Figure supplement 1.** No evidence for a relationship between Archaea-Bacteria (AB) branch length and gene evolutionary rate (average MAD root-to-tip distance).

**Figure supplement 2.** No evidence for a relationship between Archaea-Bacteria (AB) branch length and gene evolutionary rate (average MAD root-to-tip distance).

**Figure supplement 3.** A significant positive relationship between relative Archaea-Bacteria (AB) distance and evolutionary rate (MAD root-to-tip distance).

**Figure supplement 4.** Two proxies for marker gene verticality, ΔLL and between-domain split score, are highly correlated.

**Figure supplement 5.** Low-verticality genes (as measured by ΔLL) have a higher evolutionary rate (as measured by the mean root-to-tip distance on MAD-rooted gene trees).

**Figure supplement 6.** Low-verticality genes (measured by between-domain split score) have a higher evolutionary rate (MAD root-to-tip distance).

**Figure supplement 7.** High-verticality marker genes have longer Archaea-Bacteria (AB) branch lengths.

**Figure supplement 8.** Archaea-Bacteria (AB) branch length and relative AB distance are positively correlated.

**Figure supplement 9.** Archaea-Bacteria (AB) branch length is negatively correlated with between-domain split score.

**Figure supplement 10.** Within-domain split score and ΔLL are strongly correlated, suggesting that both proxies capture a common signal of marker gene verticality.

**Figure supplement 11.** Low-verticality marker genes (measured as within-domain split score) have shorter relative Archaea-Bacteria (AB) distances.

**Figure supplement 12.** Low-verticality marker genes (measured as within-domain split score) have shorter Archaea-Bacteria (AB) branch lengths.

**Figure supplement 13.** Low-verticality marker genes (measured as within-domain split score) have shorter Archaea-Bacteria (AB) branch lengths.

**Figure supplement 14.** Raw count and percentage distribution of the Genome Taxonomy Database (GTDB)-defined classes for 10,575 archaeal and bacterial genomes in the expanded marker set analysis.

**Figure supplement 15.** Raw count and percentage distribution of the Genome Taxonomy Database (GTDB)-defined phyla for 10,575 archaeal and bacterial genomes in the expanded marker set analysis.

**Figure supplement 16.** Raw count and percentage distribution of domains for 10,575 archaeal and bacterial genomes in the expanded marker set analysis.

encoding ribosomal proteins; *Petitjean et al., 2014*); and (iv) the bacterial set (29 genes used in a recent analysis of bacterial phylogeny; *Coleman et al., 2021*).

To investigate why 151 of the marker genes rejected the reciprocal monophyly of Archaea and Bacteria, we returned to the full dataset (*Zhu et al., 2019*), annotated each sequence in each marker gene family by assigning proteins to KOs, PFAMs and Interpro domains, among others (*Supplementary file 1*, see Materials and methods for details), and manually inspected the tree topologies (*Supplementary file 1*). This revealed that the major cause of domain polyphyly observed in gene trees was inter-domain gene transfer (in 359 out of 381 gene trees [94.2%]) and mixing of sequences from distinct paralogous families (in 246 out of 381 gene trees [64.6%]). For instance, marker genes encoding ABC-type transporters (p0131, p0151, p0159, p0174, p0181, p0287, p0306, p0364), tRNA synthetases (i.e., p0000, p0011, p0020, p0091, p0094, p0202), and aminotransferases and

dehydratases (i.e., p0073/4-aminobutyrate aminotransferase; p0093/3-isopropylmalate dehydratase) often comprised a mixture of paralogs.

Together, these analyses indicate that the evolutionary histories of the individual markers of the expanded set differ from each other and from the species tree. The original study investigated and acknowledged (*Zhu et al., 2019*) the varying levels of congruence between the marker phylogenies and the species tree, but did not investigate the underlying causes. Our analyses establish the basis for these disagreements in terms of gene transfers and the mixing of orthologs and paralogs within and between domains. The estimation of genetic distance based on concatenation relies on the assumption that all of the genes in the supermatrix evolve on the same underlying tree; genes with different gene tree topologies violate this assumption and should not be concatenated because the topological differences among sites are not modeled, and so the impact on inferred branch lengths is difficult to predict. In practice, it is often difficult to be certain that all of the markers in a concatenate share the same gene tree topology, and the analysis proceeds on the hypothesis that a small proportion of discordant genes are not expected to seriously impact the inferred tree. However, the concatenated tree inferred from the expanded marker set differs from previous trees in that the genetic distance between Bacteria and Archaea is greatly reduced, such that the AB branch length appears comparable to distances among bacterial phyla (*Zhu et al., 2019*). Since an accurate estimate of the AB branch length has a major bearing on unanswered questions regarding the root of the universal tree (*Gouy et al., 2015*), we next evaluated the impact of the conflicting gene histories within the expanded marker set on inferred AB branch length.

## The inferred branch length between Archaea and Bacteria is shortened by inter-domain gene transfer and hidden paralogy

To investigate the impact of gene transfers and mixed paralogy on the AB branch length inferred by gene concatenations (*Zhu et al., 2019*), we compared branch lengths estimated from markers on the basis of whether or not they rejected domain monophyly in the expanded marker set (*Figure 1A*). To estimate AB branch lengths for genes in which the domains were not monophyletic in the ML tree, we first performed a constrained ML search to find the best gene tree that was consistent with domain monophyly for each family under the LG + G4 + F model in IQ-TREE 2 (*Minh et al., 2020*). While it may seem strained to estimate the length of a branch that does not appear in the ML tree, we reasoned that this approach would provide insight into the contribution of these genes to the AB branch length in the concatenation, in which they conflict with the overall topology. AB branch lengths were significantly ($p=3.653 \times 10^{-6}$, Wilcoxon rank-sum test) shorter for markers that rejected domain monophyly (Bonferroni-corrected $p<0.0001656$; *Figure 1A*): the mean AB branch length was 0.00668 substitutions/site for markers that significantly rejected domain monophyly and 0.287 substitutions/site for markers that did not reject domain monophyly. This behavior might result from marker gene transfers reducing the number of fixed differences between the domains, so that the AB branch length in a tree in which Archaea and Bacteria are constrained to be reciprocally monophyletic will tend towards 0 as the number of transfers increases.

To test the hypothesis that phylogenetic incongruence among markers might reduce the inferred AB distance, we evaluated the relationship between AB distance and two complementary metrics of marker gene verticality: ΔLL, the difference in log-likelihood between the constrained ML tree and the ML gene tree (a proxy for the extent to which a marker gene rejects the reciprocal monophyly of Bacteria and Archaea), and the 'split score' (*Dombrowski et al., 2020*), which measures the extent to which marker genes recover established relationships for defined taxonomic levels of interest (e.g., at the level of domain, phylum, or order), averaging over bootstrap distributions of gene trees to account for phylogenetic uncertainty (see Materials and methods). We evaluated split scores at both the between-domain and within-domain (*Figure 1—figure supplements 1–13*) levels. ΔLL and between-domain split score were positively correlated with each other (*Figure 1—figure supplement 4*) and negatively correlated with both AB stem length (*Figure 1C and E*) and relative AB distance (*Figure 1D and F*), an alternative metric (*Zhu et al., 2019*) that compares average tip-to-tip distances within and between domains. Interestingly, between-domain and within-domain split scores were strongly positively correlated (*Figure 1G*), and the same relationships between within-domain split score, AB branch length, and relative AB distance were observed (*Figure 1—figure supplements 11 and 12*). Overall, these results suggest that genes that recover the reciprocal monophyly of Archaea

and Bacteria also evolve more vertically within each domain, and that these vertically evolving marker genes support a longer AB branch and a greater AB distance. Indeed, *Zhu et al., 2019* also recovered a significant positive relationship between gene verticality and relative AB distance (see their Fig. 5E). Consistent with these inferences, AB branch lengths estimated using concatenation decreased as increasing numbers of low-verticality markers (i.e., markers with higher ΔLL) were added to the concatenate (*Figure 1*). These results suggest that inter-domain gene transfers reduce the overall AB branch length when included in a concatenation.

An alternative explanation for the positive relationship between marker gene verticality and AB branch length could be that vertically evolving genes experience higher rates of sequence evolution. For a set of genes that originate at the same point on the species tree, the mean root-to-tip distance (measured in substitutions per site, for gene trees rooted using the MAD (minimal ancestor deviation) method; *Tria et al., 2017*) provides a proxy of evolutionary rate. Mean root-to-tip distances were significantly positively correlated with ΔLL and between-domain split score (ΔLL: $R = 0.1397803$, p=0.01506, split score: $R = 0.1705415$, p=0.002947; *Figure 1—figure supplements 5 and 6*), indicating that vertically evolving genes evolve relatively slowly (note that large values of ΔLL and split score denote low verticality). Thus, the longer AB branches of vertically evolving genes do not appear to result from a faster evolutionary rate for these genes. Taken together, these results indicate that the inclusion of genes that do not support the reciprocal monophyly of Archaea and Bacteria, or their constituent taxonomic ranks, in the universal concatenate explains the reduced estimated AB branch length.

## Finding ancient vertically evolving genes

To estimate the AB branch length and the phylogeny of prokaryotes using a dataset that resolves some of the issues identified above, we performed a meta-analysis of several previous studies to identify a consensus set of vertically evolving marker genes. We identified unique markers from these analyses by reference to the COG ontology (*Supplementary file 2*, *Dombrowski et al., 2020*; *Galperin et al., 2019*), extracted homologous sequences from a representative sample of 350 archaeal and 350 bacterial genomes (*Supplementary file 3*), and performed iterative phylogenetics and manual curation to obtain a set of 54 markers that recovered archaeal and bacterial monophyly (see Materials and methods). Prior to manual curation, non-ribosomal markers had a greater number of HGTs (horizontal gene transfer) and cases of mixed paralogy. In particular, for the original set of 95 unique COG families (see 'Phylogenetic analyses' in Materials and methods), we rejected 41 families based on the inferred ML trees either due to a large degree of HGT, paralogous gene families, or LBA (long branch attraction). For the remaining 54 markers, the ML trees contained evidence of occasional recent HGT events. Strict monophyly was violated in 69% of the non-ribosomal and 29% of the ribosomal families. We manually removed the individual sequences that violated domain monophyly before realignment, trimming, and subsequent tree inference (see Materials and methods). These results imply that manual curation of marker genes is important for deep phylogenetic analyses, particularly when using non-ribosomal markers. Comparison of within-domain split scores for these 54 markers (*Supplementary file 4*) indicated that markers that better resolved established relationships within each domain also supported a longer AB branch length (*Figure 2A*). Further, the AB branch length inferred from a concatenation of the 54 marker genes increased moderately following pruning of recent HGTs, from 1.734 substitutions/site (non-pruned) to 1.945 substitutions/site after manual pruning, consistent with the hypothesis that non-modeled inter-domain HGTs reduce the overall estimate of AB branch length when included in concatenations.

## Distributions of AB branch lengths for ribosomal and non-ribosomal marker genes are similar

Traditional universal marker sets include many ribosomal proteins (*Ciccarelli et al., 2006*; *Fournier and Gogarten, 2010*; *Harris et al., 2003*; *Hug et al., 2016*; *Liu et al., 2021*; *Williams et al., 2020*). If ribosomal proteins experienced accelerated evolution during the divergence of Archaea and Bacteria, this might lead to the inference of an artifactually long AB branch length (*Petitjean et al., 2014*; *Zhu et al., 2019*). To investigate this, we plotted the inter-domain branch lengths for the 38 and 16 ribosomal and non-ribosomal genes, respectively, comprising the 54 marker genes set. We found no evidence that there was a longer AB branch associated with ribosomal markers than for other

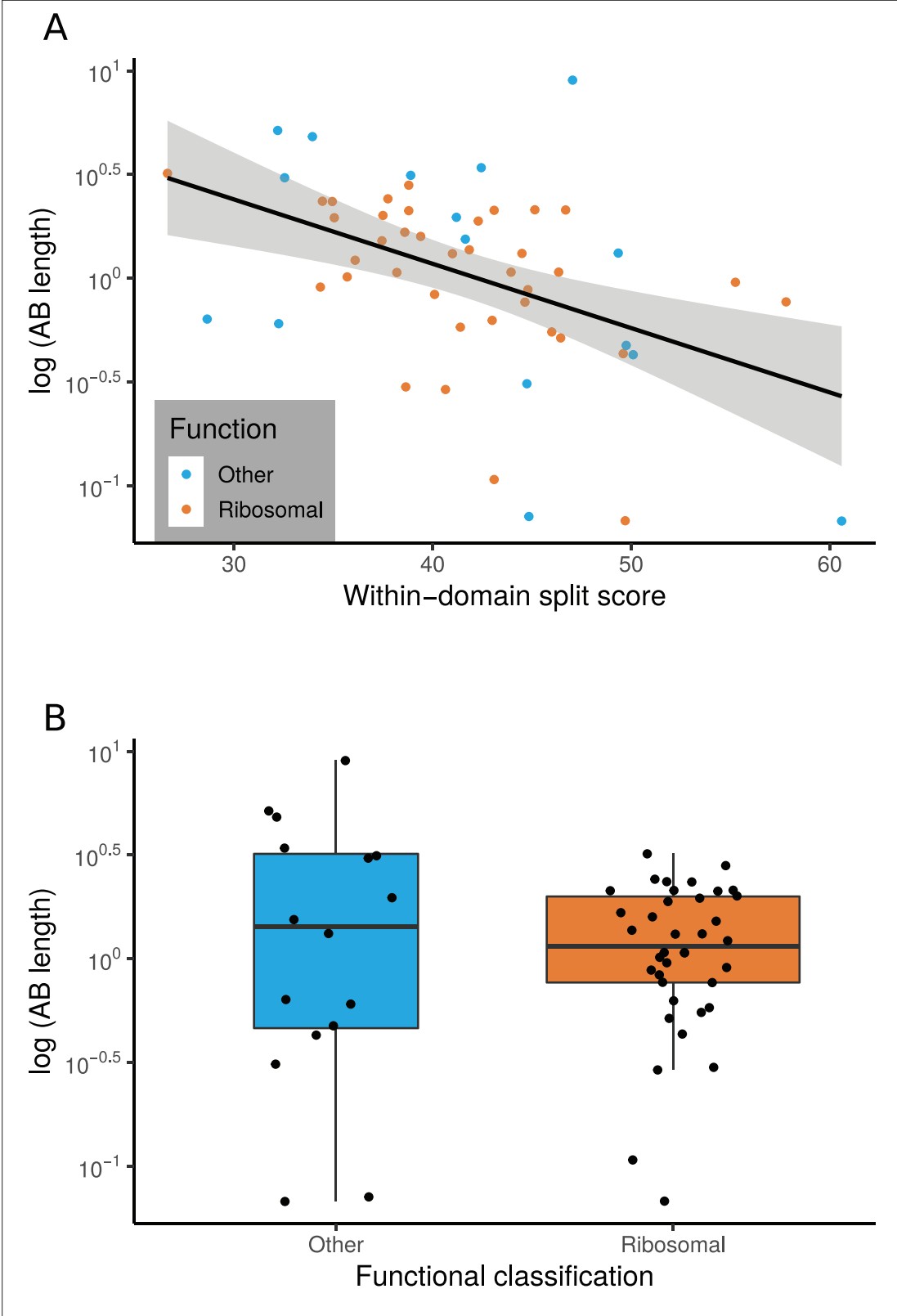

**Figure 2.** The relationship between marker gene verticality, Archaea-Bacteria (AB) branch length, and functional category. (**A**) Vertically evolving phylogenetic markers have longer AB branches. The plot shows the relationship between a proxy for marker gene verticality, within-domain split score (a lower split score denotes better recovery of established within-domain relationships, see Materials and methods), and AB branch length (in expected number of substitutions/site) for the 54 marker genes. Marker genes with higher split scores (that split established monophyletic groups

*Figure 2 continued on next page*

*Figure 2 continued*

into multiple subclades) have shorter AB branch lengths (p=0.0311, *R* = 0.294). Split scores of ribosomal and non-ribosomal markers were statistically indistinguishable (p=0.828, ***Figure 2—figure supplement 1***). (**B**) Among vertically evolving marker genes, ribosomal genes do not have a longer AB branch length. The plot shows functional classification of markers against AB branch length using 54 vertically evolving markers. We did not obtain a significant difference between AB branch lengths for ribosomal and non-ribosomal genes (p=0.6191, Wilcoxon rank-sum test).

The online version of this article includes the following figure supplement(s) for figure 2:

**Figure supplement 1.** Among vertically evolving marker genes, the split scores of ribosomal and non-ribosomal proteins are statistically indistinguishable.

vertically evolving 'core' genes (***Figure 2B***; mean AB branch length for ribosomal proteins 1.35 substitutions/site, mean for non-ribosomal 2.25 substitutions/site). To investigate further, we compared AB branch lengths inferred from concatenates of the ribosomal and non-ribosomal subsets of the 54 ancient, vertically evolving genes (***Table 1***). AB branch lengths from the ribosomal and non-ribosomal concatenates were similar overall, with some support for a longer AB branch length from vertically evolving non-ribosomal genes. Thus, these data do not support an accelerated evolutionary rate for ribosomal genes compared to other kinds of genes on the AB branch.

## Substitutional saturation and poor model fit contribute to underestimation of AB branch length

For the 27 most vertically evolving genes as ranked by within-domain split score, we performed an additional round of single-gene tree inference and manual review to identify and remove the remaining sequences that had evidence of HGT or represented distant paralogs. The resulting single-gene trees are provided in the Data Supplement (10.6084/m9.figshare.13395470). To evaluate the relationship between site evolutionary rate and AB branch length, we created two concatenations: fastest sites (comprising sites with the highest probability of being in the fastest gamma rate category; 868 sites) and slowest sites (sites with the highest probability of being in the slowest gamma rate category, 1604 sites) and compared relative branch lengths inferred from the entire concatenate using IQ-TREE 2 to infer site-specific rates (***Figure 3***).

Notably, the proportion of inferred substitutions that occur along the AB branch differs between the slow-evolving and fast-evolving sites. As would be expected, the total tree length measured in substitutions per site is shorter from the slow-evolving sites, but the relative AB branch length is longer (1.2 substitutions/site, or ~2% of all inferred substitutions, compared to 2.6 substitutions/site, or ~0.04% of all inferred substitutions for the fastest-evolving sites; see ***Figure 3—figure supplement 1*** for absolute tree size comparisons). Since we would not expect the distribution of substitutions over the tree to differ between slow-evolving and fast-evolving sites, this result suggests that some ancient changes along the AB branch at fast-evolving sites have been overwritten by more recent events in evolution – that is, that substitutional saturation leads to an underestimate of the AB branch length (this is the case for both the expanded marker set and the 27 marker set; ***Figure 3—figure supplement 2***).

Another factor that has been shown to lead to underestimation of genetic distance on deep branches is a failure to adequately model the site-specific features of sequence evolution (***Lartillot and Philippe, 2004***; ***Schrempf et al., 2020***; ***Wang et al., 2018***; ***Williams et al., 2020***; ***Zhu et al.,***

**Table 1.** Archaea-Bacteria (AB) branch lengths and AB branch lengths as a proportion of total tree length inferred from ribosomal and non-ribosomal concatenates are similar.
The data do not support a faster evolutionary rate for ribosomal proteins on the AB branch compared to other kinds of ancient proteins.

|  | AB branch length | | Total tree length | | AB branch length as a proportion of total tree length | |
|---|---|---|---|---|---|---|
|  | Ribosomal | Non-ribosomal | Ribosomal | Non-ribosomal | Ribosomal | Non-ribosomal |
| 27 marker set | 1.9541 | 3.7723 | 250.7255 | 239.8203 | 0.0078 | 0.0157 |
| 54 marker set | 1.8647 | 2.5414 | 271.3327 | 288.8470 | 0.0069 | 0.0088 |

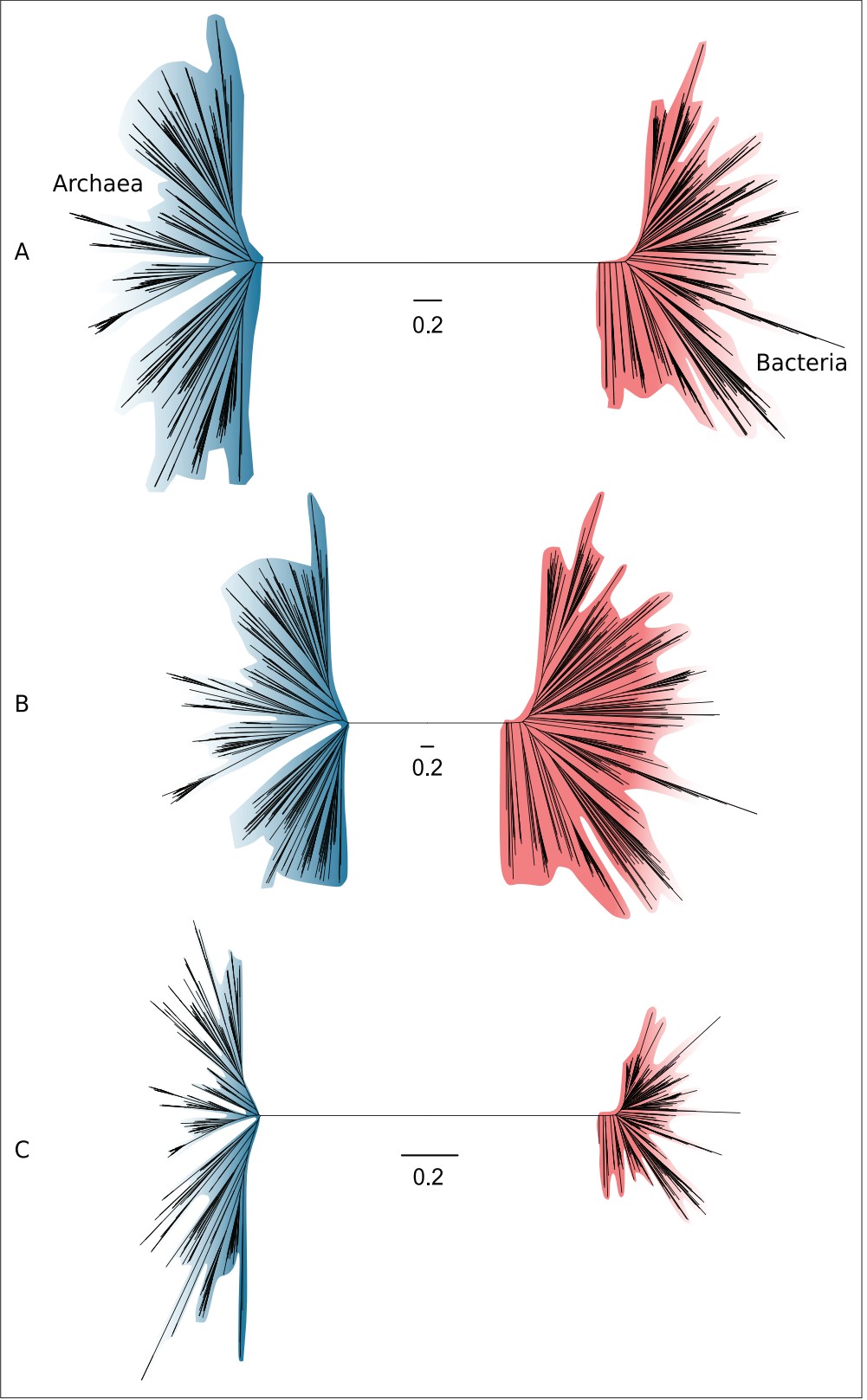

**Figure 3.** Slow- and fast-evolving sites support different shapes for the universal tree. (**A**) Tree of Archaea (blue) and Bacteria (red) inferred from a concatenation of 27 core genes using the best-fitting model (LG + C60 + G4 + F). (**B**) Tree inferred from the fastest-evolving sites. (**C**) Tree inferred from the slowest-evolving sites. To facilitate comparison of relative diversity, scale bars are provided separately for each panel; for a version of this figure

*Figure 3 continued on next page*

*Figure 3 continued*

with a common scale bar for all three panels, see *Figure 3—figure supplement 1*. Slow-evolving sites support a relatively long inter-domain branch and less diversity within the domains (i.e., shorter between-taxa branch lengths within domains). This suggests that substitution saturation (overwriting of earlier changes) may reduce the relative length of the AB branch at fast-evolving sites and genes.

The online version of this article includes the following figure supplement(s) for figure 3:

**Figure supplement 1.** Slow- and fast-evolving sites support different shapes for the universal tree.

**Figure supplement 2.** Vertically evolving genes and slow-evolving sites support a longer relative Archaea-Bacteria (AB) branch length.

**Figure supplement 3.** The effect of modeling site compositional heterogeneity on Archaea-Bacteria (AB) branch length.

---

*2019*). Amino acid preferences vary across the sites of a sequence alignment due to variation in the underlying functional constraints (*Lartillot and Philippe, 2004*; *Quang et al., 2008*; *Wang et al., 2008*). The consequence is that, at many alignment sites, only a subset of the 20 possible amino acids are tolerated by selection. Standard substitution models such as LG + G4 + F are site-homogeneous and approximate the composition of all sites using the average composition across the entire alignment. Such models underestimate the rate of evolution at highly constrained sites because they do not account for the high number of multiple substitutions that occur at such sites. The effect is that site-homogeneous models underestimate branch lengths when fit to site-heterogeneous data. Site-heterogeneous models have been developed that account for site-specific amino acid preferences, and these generally show improved fit to real protein sequence data (reviewed in *Williams et al., 2021*). To evaluate the impact of substitution models on estimates of AB branch length, we assessed the fit of a range of models to the full concatenation using the Bayesian information criterion (BIC) in IQ-TREE 2. The AB branch length inferred under the best-fit model, the site-heterogeneous LG + C60 + G4 + F model, was 2.52 substitutions/site, ~1.7-fold greater than the branch length inferred from the site-homogeneous LG + G4 + F model (1.45 substitutions/site). Thus, substitution model fit has a major effect on the estimated length of the AB branch, with better-fitting models supporting a longer branch length (*Table 2*). The same trends are evident when better-fitting site-heterogeneous models are used to analyze the expanded marker set: considering only the top 5% of genes by ΔLL score, the AB branch length is 1.2 under LG + G4 + F, but increases to 2.4 under the best-fitting LG + C60 + G4 + F model (*Figure 3—figure supplement 3*). These results are consistent with *Zhu et al., 2019*, who also noted that AB branch length increases as model fit improves for the expanded marker dataset.

Overall, these results indicate that difficulties with modeling sequence evolution, either due to substitutional saturation or failure to model variation in site compositions, lead to an underestimation of the AB branch length, both in published analyses and for the analyses of the new dataset presented here. As substitution models improve, we would therefore expect estimates of the AB branch length to increase further.

## A phylogeny of Archaea and Bacteria inferred from 27 vertically evolving marker genes

The phylogeny of the primary domains of life inferred from the 27 most vertically evolving genes as inferred based on our ranking of markers and using the best-fitting LG + C60 + G4 + F model (*Figure 4*) is consistent with recent single-domain trees inferred for Archaea and Bacteria independently (*Coleman et al., 2021*; *Dombrowski et al., 2020*; *Williams et al., 2017*), although the deep relationships within Bacteria are poorly resolved, with the exception of the monophyly of Gracilicutes (*Figure 4*). Our results are also in

**Table 2.** The inferred Archaea-Bacteria (AB) branch length from a concatenation of the top 27 markers using a simple model compared to models that account for site compositional heterogeneity.

Models that account for across-site compositional heterogeneity fit the data better (as assessed by lower Bayesian information criterion [BIC] scores) and infer a longer AB branch length.

| Substitution model | BIC (ΔBIC) | AB branch length |
|---|---|---|
| LG + G4 + F | 5935950.053 | 1.4491 |
| LG + C20 + G4 + F | (152046.1) | 2.1394 |
| LG + C40 + G4 + F | (179126.7) | 2.4697 |
| LG + C60 + G4 + F | (189063.8) | 2.5178 |

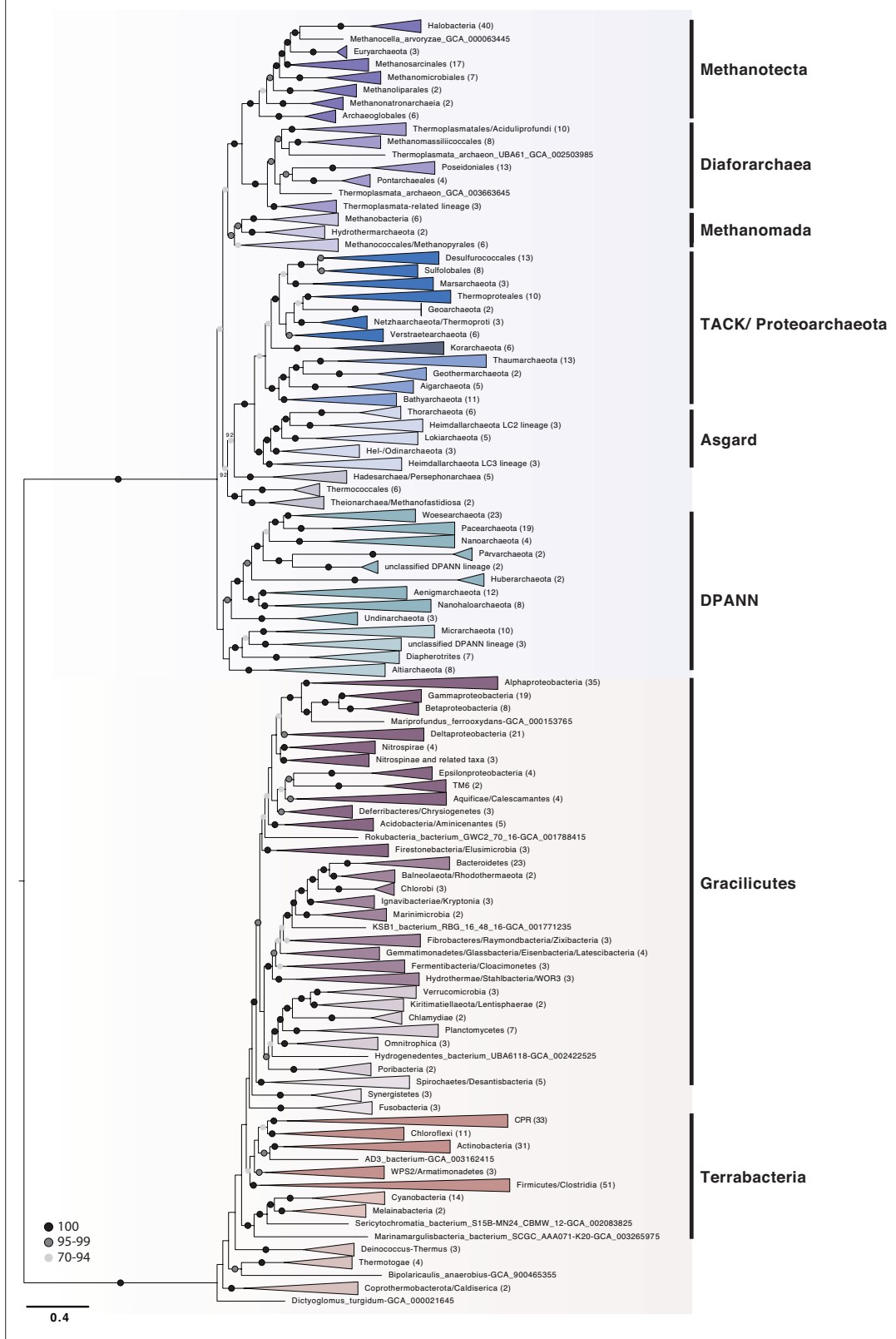

**Figure 4.** A phylogeny of Archaea and Bacteria inferred from a concatenation of 27 marker genes. Consistent with some recent studies (*Dombrowski et al., 2020*; *Guy and Ettema, 2011*; *Raymann et al., 2015*; *Williams et al., 2017*), we recovered the DPANN, TACK, and Asgard Archaea as monophyletic groups. Although the deep branches within Bacteria are poorly resolved, we recovered a sister group relationship between candidate

*Figure 4 continued on next page*

*Figure 4 continued*

phyla radiation (CPR) and Chloroflexota, consistent with recent reports (*Taib et al., 2020*; *Coleman et al., 2021*). The tree was inferred using the best-fitting LG + C60 + G4 + F model in IQ-TREE 2 (*Minh et al., 2020*). Branch lengths are proportional to the expected number of substitutions per site. Support values are ultrafast (UFBoot2) bootstraps (*Hoang et al., 2018*). Numbers in parenthesis refer to the number of taxa within each collapsed clade. Please note that the collapsed taxa in the Archaea and Bacteria roughly correspond to order- and phylum-level lineages, respectively.

The online version of this article includes the following figure supplement(s) for figure 4:

**Figure supplement 1.** Raw count and percentage distribution of the Genome Taxonomy Database (GTDB)-defined classes of 350 archaea and 350 bacteria used in the 27 marker gene set analysis.

**Figure supplement 2.** Raw count and percentage distribution of the Genome Taxonomy Database (GTDB)-defined phyla of 350 archaea and 350 bacteria used in the 27 marker gene set analysis.

**Figure supplement 3.** A phylogeny of Archaea and Bacteria inferred from a concatenation of 25 marker genes.

good agreement with a recent estimate of the universal tree based on a different marker gene selection approach (*Martinez-Gutierrez and Aylward, 2021*). In that study, marker genes were selected based on Tree Certainty, a metric that quantifies phylogenetic signal based on the extent to which markers distinguish between different resolutions of conflicting relationships (*Salichos and Rokas, 2013*).

In particular, our analysis placed the candidate phyla radiation (CPR) (*Brown et al., 2015*) as a sister lineage to Chloroflexi (Chloroflexota) rather than as a deep-branching bacterial superphylum. While this contrasts with initial trees suggesting that CPR may represent an early diverging sister lineage of all other Bacteria (*Brown et al., 2015*; *Castelle and Banfield, 2018*; *Hug et al., 2016*), our finding is consistent with recent analyses that have instead recovered CPR within the Terrabacteria (*Coleman et al., 2021*; *Martinez-Gutierrez and Aylward, 2021*; *Taib et al., 2020*). Together, these analyses suggest that the deep-branching position of CPR in some trees may be a result of long branch attraction, a possibility that has been raised previously (*Hug et al., 2016*; *Méheust et al., 2019*).

The deep branches of the archaeal subtree are generally well-resolved and recover DPANN (51% bootstrap support), Asgards (100% bootstrap support), and TACK Archaea (75% bootstrap support) as monophyletic clades in agreement with a range of previous studies (*Guy and Ettema, 2011*; *Dombrowski et al., 2020*; *Guy and Ettema, 2011*; *Raymann et al., 2015*; *Williams et al., 2017*). We also find support for the placement of Methanonatronarchaeia (*Sorokin et al., 2017*) distant to Halobacteria as one of the earliest branches of the Methanotecta (*Figure 4*, *Figure 4—figure supplement 3*) in agreement with recent analyses, suggesting that their initial placement with Halobacteria (*Sorokin et al., 2017*) may be an artifact of compositional attraction (*Aouad et al., 2019*; *Dombrowski et al., 2020*; *Feng et al., 2021*; *Martijn et al., 2020*).

We obtained moderate (92%) bootstrap support for the branching of some Euryarchaeota with the TACK + Asgard clade: the Hadesarchaea + Persephonarchaea were resolved as the sister group to TACK + Asgards with moderate (92%) support, with this entire lineage branching sister to a strongly supported (100%) clade comprising Theionarchaea, Methanofastidiosa, and Thermococcales. However, the position of these lineages was sensitive to the marker gene set used. As part of a robustness test, we also inferred an additional tree from a 25-gene subset, excluding two genes that have complex evolutionary histories in Archaea (*Narrowe et al., 2018*; *Figure 4—figure supplement 3*). In this analysis, these Archaea instead branched with Methanomada with high support (98%), highlighting the difficulty of placing these lineages in the archaeal tree. Euryarchaeotal paraphyly has been previously reported (*Adam et al., 2017*; *Raymann et al., 2015*; *Williams et al., 2017*; *Aouad et al., 2022*), though the extent of the observed paraphyly and the lineages involved has varied among analyses.

A basal placement of DPANN within Archaea is sometimes viewed with suspicion (*Aouad et al., 2018*) because DPANN genomes are reduced and appear to be fast-evolving, properties that may cause LBA artifacts (*Dombrowski et al., 2019*) when analyses include Bacteria. However, in contrast to CPR, with which DPANN share certain ecological and genomic similarities (e.g., host dependency, small genomes, limited metabolic potential), the early divergence of DPANN from the archaeal branch has received support from a number of recent studies (*Baker et al., 2020*; *Beam et al., 2020*; *Dombrowski et al., 2020*; *Rinke et al., 2021*; *Williams et al., 2017*; *Zaremba-Niedzwiedzka et al.,*

*2017*; *Aouad et al., 2022*), though the inclusion of certain lineages within this radiation remains controversial (*Aouad et al., 2018*; *Feng et al., 2021*) and the placement of the root is uncertain (*Dombrowski et al., 2020*; *Aouad et al., 2022*). While more in-depth analyses will be needed to further illuminate the evolutionary history of DPANN and establish whether the group as a whole is monophyletic, our work is in agreement with current literature and a recently established phylogeny-informed archaeal taxonomy (*Rinke et al., 2021*).

A broader observation from our analysis is that the phylogenetic diversity of the archaeal and bacterial domains, measured as substitutions per site in this consensus set of vertically evolving marker genes, appears to be similar (*Figure 3A*; the mean root-to-tip distance for archaea: 2.38, for bacteria: 2.41; the range of root-to-tip distances for archaea: 1.79–3.01, for bacteria: 1.70–3.17). Considering only the slowest-evolving category of sites, branch lengths within Archaea are actually longer than within Bacteria (*Figure 3C*). This result differs from some published trees (*Hug et al., 2016*; *Zhu et al., 2019*) in which the phylogenetic diversity of Bacteria has appeared to be significantly greater than that of Archaea. By contrast to those earlier studies, we analyzed a set of 350 genomes from each domain, an approach that may tend to reduce the differences between them. While we had to significantly downsample the sequenced diversity of Bacteria, our sampling nonetheless included representatives from all known major lineages of both domains (*Figure 4—figure supplements 1 and 2*, see *Figure 1—figure supplements 14–16* for a comparison with the expanded marker set), and so might be expected to recover a difference in diversity, if present. Our analyses and a number of previous studies (*Hug et al., 2016*; *Parks et al., 2018*; *Petitjean et al., 2014*; *Zhu et al., 2019*) indicate that the choice of marker genes has a profound impact on the apparent phylogenetic diversity of certain prokaryotic groups; for instance, in the proportion of bacterial diversity composed of CPR (*Hug et al., 2016*; *Parks et al., 2017*). Our results demonstrate that slow- and fast-evolving sites from the same set of marker genes support different tree shapes and branch lengths; it therefore seems possible that between-dataset differences are due, at least in part, to evolutionary rate variation within and between marker genes.

## Difficulties in estimating the age of the last universal common ancestor

While a consensus may be emerging on the topology of the universal tree, estimates of the ages of the deepest branches, and their lengths in geological time, remain highly uncertain. The fossil record of early life is incomplete and difficult to interpret (*Wacey, 2009*), and in this context molecular clock methods provide a means of combining the abundant genetic data available for modern organisms with the limited fossil record to improve our understanding of early evolution (*Betts et al., 2018*). The 381-gene dataset was suggested to be (*Zhu et al., 2019*) useful for inferring deep divergence times because age estimates of LUCA (last universal common ancestor) from this dataset using a strict molecular clock were in agreement with the geological record: a root (LUCA) age of 3.6–4.2 Ga was inferred from the entire 381-gene dataset, consistent with the earliest fossil evidence for life (*Betts et al., 2018*; *Sugitani et al., 2015*). By contrast, analysis of ribosomal markers alone (*Zhu et al., 2019*) supported a root age of ~7 Ga, which might be considered implausible because it is older than the age of the Earth and Solar System (with the moon-forming impact occurring ~4.52 Ga; *Barboni et al., 2017*; *Hanan and Tilton, 1987*).

The published molecular clock analyses (*Zhu et al., 2019*) made use of concatenation-based branch lengths in which topological disagreement among sites is not modeled and are likely to be affected by the impact of nonvertical marker genes and substitutional saturation on branch length estimation discussed above. Consistent with this hypothesis, divergence time inference using the same method on the 5% most-vertical subset of the expanded marker set (as determined by ΔLL; this set of 20 genes includes only one ribosomal protein, see *Supplementary file 5a*) resulted in age estimates for LUCA that exceed the age of the Earth, 5.6–6.15 Ga (*Figure 5*), approaching the age inferred from the ribosomal genes (7.46–8.03 Ga). These results (*Figure 5*) suggest that the apparent agreement between the fossil record and divergence times estimated from the expanded gene set may be due, at least in part, to the shortening of the AB branch due to phylogenetic incongruence among marker genes.

In the original analyses, the age of LUCA was estimated using a strict clock with a single calibration constraining the split between Cyanobacteria and Melainabacteria derived from estimates of the Great Oxidation Event and a secondary estimate of the age of cyanobacteria derived from an

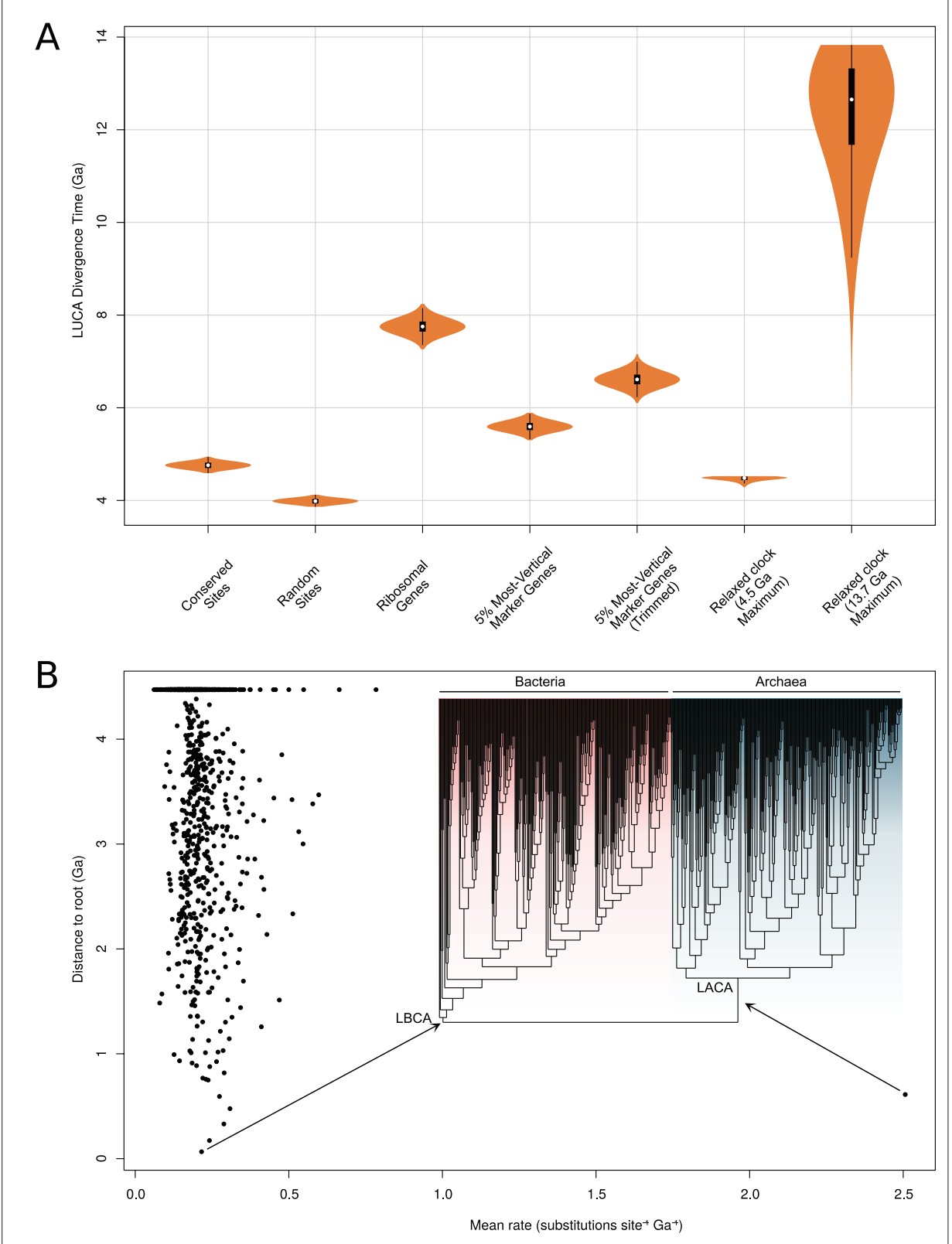

**Figure 5.** Molecular clock estimates of the LUCA and LACA age are uncertain due to a lack of deep calibrations and maximum ages for microbial clades. (**A**) Posterior node age estimates from Bayesian molecular clock analyses of (1) conserved sites as estimated previously (*Zhu et al., 2019*); (2) random sites (*Zhu et al., 2019*); (3) ribosomal genes (*Zhu et al., 2019*); (4) the top 5% of marker gene families according to their ΔLL score (including only one ribosomal protein); and (5) the same top 5% of marker genes trimmed using BMGE (*Criscuolo and Gribaldo, 2010*) to remove poorly aligned

*Figure 5 continued on next page*

Figure 5 continued

sites. In each case, a strict molecular clock was applied, with the age of the Cyanobacteria-Melainabacteria split constrained between 2.5 and 2.6 Ga. In (6) all annotations for the top 20 genes and (7) an expanded set of fossil calibrations was implemented with a relaxed (lognormal) molecular clock. In (6), a soft maximum age of 4.520 Ga was applied, representing the age of the moon-forming impact (*Kleine et al., 2005*). In (7), a soft maximum age corresponding to the estimated age of the universe (*Aghanim, 2018*) was applied. (**B**)Inferred rates of molecular evolution along the phylogeny in a relaxed clock analysis where the maximum age was set to 4.520 Ga. We plotted per-branch rates (site$^{-1}$ Ga$^{-1}$) against distance to the root. The rate of evolution along the archaea stem lineage was a clear outlier (mean = 2.51, 95% HPD = 1.6–3.5subs. site$^{-1}$ Ga$^{-1}$). The phylogeny used was that depicted in *Figure 4*.

independent analysis (*Shih et al., 2017*). The combination of a strict clock and only two calibrations is not sufficient to capture the variation in evolutionary rate over deep timescales (*Drummond et al., 2006*). To investigate whether additional calibrations might help to improve age estimates for deep nodes in the universal tree, we performed analyses on our new 27 marker gene dataset using two different relaxed clock models (with branchwise independent and autocorrelated rates) and seven additional calibrations (*Supplementary file 5b*). Unfortunately, all of these were minimum age calibrations with the exception of the root (for which the moon-forming impact 4.52 Ga [*Kleine et al., 2005*] provides a reasonable maximum) due to the difficulty of establishing uncontroversial maximum ages for microbial clades. Maximum age constraints are essential to inform faster rates of evolution because, in combination with more abundant minimum age constraints, they imply that a given number of substitutions must have accumulated in at most a certain interval of time. In the absence of other maximum age constraints, the only lower bound on the rate of molecular evolution is provided by the maximum age constraint on the root (LUCA).

These new analyses indicated that even with additional minimum age calibrations the age of LUCA inferred from the 27-gene dataset was unrealistically old, falling close to the maximum age constraint in all analyses even when the maximum was set to the age of the known universe (13.7 Ga; *Aghanim, 2018*; *Figure 5*). Inspection of the inferred rates of molecular evolution across the tree (*Figure 5B*) provides some insight into these results: the mean rate is low (mean = 0.21, 95% credibility interval = 0.19–0.22 subs. site$^{-1}$ Ga$^{-1}$), so that long branches (such as the AB stem), in the absence of other information, are interpreted as evidence of a long period of geological time. These low rates likely result both from the limited number of calibrations and, in particular, the lack of maximum age constraints.

An interesting outlier among inferred rates is the LUCA to LACA (last archaeal common ancestor) branch, which has a rate 10-fold greater than the average (mean = 2.51, 95% HPD (highest posterior density) = 1.6–3.5 subs. site$^{-1}$ Ga$^{-1}$). The reason is that calibrations within Bacteria imply that LBCA (last bacterial common ancestor) cannot be younger than 3.225 Ga (Manzimnyama Banded Ironstone Formation provides evidence of cyanobacterial oxygenation; *Satkoski et al., 2015*, *Supplementary file 5b*); as a result, with a 4.52 Ga maximum, the LUCA to LBCA branch cannot be longer than 1.295 Ga. By contrast, the early branches of the archaeal tree are poorly constrained by fossil evidence. Analysis without the 3.225 Ga constraint resulted in overlapping age estimates for LBCA (4.47–3.53 Ga) and LACA (4.37–3.44 Ga). Finally, analysis of the archaeal and bacterial subtrees independently (i.e., without the AB branch, rooted on LACA and LBCA, respectively) resulted in LBCA and LACA ages that abut the maximum root age (LBCA: 4.52–4.38 Ga; LACA: 4.52–4.14 Ga). This analysis demonstrates that, under these analysis conditions, the inferred age of the root (whether corresponding to LUCA, LACA, or LBCA) is strongly influenced by the prior assumptions about the maximum age of the root.

In sum, the agreement between fossils and age estimates from the expanded gene set appears to result from the impact of phylogenetic incongruence on branch length estimates. Under more flexible modeling assumptions, the limitations of current clock methods for estimating the age of LUCA become manifest: the sequence data only contain limited information about the age of the root, with posterior estimates driven by the prior assumptions about the maximum age of the root. This analysis implies several possible ways to improve age estimates of deep branches in future analyses. More calibrations, particularly maximum age constraints and calibrations within Archaea, are essential to refine the current estimates. Given the difficulties in establishing maximum ages for archaeal and bacterial clades, constraints from other sources such as donor-recipient age constraints inferred from HGTs (*Davín et al., 2018*; *Fournier et al., 2021*; *Szöllősi et al., 2021*; *Wolfe and Fournier, 2018*), or clock models that capture biological opinion about rate shifts in early evolution, may be particularly valuable.

## Conclusion

Our analysis of a range of published marker gene datasets (*Petitjean et al., 2014*; *Spang et al., 2015*; *Williams et al., 2020*; *Zhu et al., 2019*) indicates that the choice of markers and the fit of the substitution model are both important for inference of deep phylogeny from concatenations, in agreement with an existing body of literature (reviewed in *Kapli et al., 2021*; *Kapli et al., 2020*; *Williams et al., 2021*). We established a set of 27 vertically evolving marker gene families and found no evidence that ribosomal genes overestimate stem length; since they appear to be transferred less frequently than other genes, our analysis affirms that ribosomal proteins are useful markers for deep phylogeny. In general, high-verticality markers, regardless of functional category, supported a longer AB branch

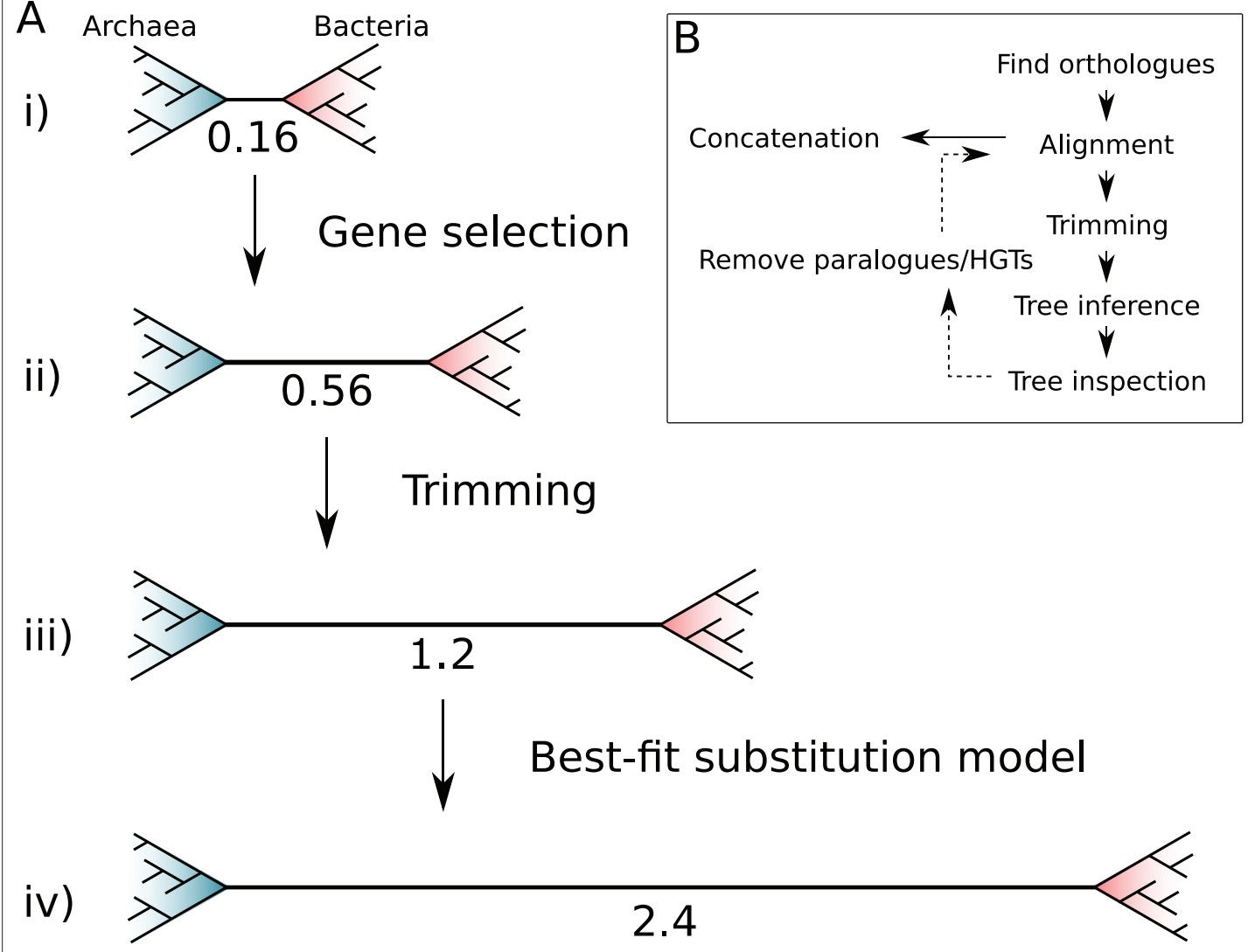

**Figure 6.** The impact of marker gene choice, phylogenetic congruence, alignment trimming, and substitution model fit on estimates of the Archaea-Bacteria (AB) branch length. (**A**) Analysis using a site-homogeneous model (LG + G4 + F) on the complete 381-gene expanded set (i) results in an AB branch substantially shorter than previous estimates. Removing the genes most seriously affected by inter-domain gene transfer (ii), trimming poorly aligned sites (iii) using BMGE (*Criscuolo and Gribaldo, 2010*) in the original alignments (see below), and using the best-fitting site-heterogeneous model (iv) (LG + C60 + G4 + F) substantially increase the estimated AB length, such that it is comparable with published estimates from the 'core' set: 3.3 (*Williams et al., 2020*) and the consensus set of 27 markers identified in the present study: 2.5. Branch lengths measured in expected number of substitutions/site. (**B**) Workflow for iterative manual curation of marker gene families for concatenation analysis. After inference and inspection of initial ortholog trees, several rounds of manual inspection and removal of HGTs and distant paralogs were carried out. These sequences were removed from the initial set of orthologs before alignment and trimming. For a detailed discussion of some of these issues, and practical guidelines on phylogenomic analysis of multi-gene datasets, see *Kapli et al., 2020* for a useful review.

length. Furthermore, our phylogeny was consistent with recent work on early prokaryotic evolution, resolving the major clades within Archaea and nesting the CPR within Terrabacteria. Notably, our analyses suggested that both the true AB branch length (*Figure 6A*) and the phylogenetic diversity of Archaea may be underestimated by even the best current models, a finding that is consistent with a root for the tree of life between the two prokaryotic domains. Taken together with fossil evidence for crown-group Bacteria ~3.2 Ga (*Betts et al., 2018*), the long AB branch length inferred from vertically evolving genes is consistent with the hypothesis that rates of molecular evolution may have been higher early in life's history than more recently (*Lopez et al., 1999*), although the inferred rates are uncertain and contingent on limited fossil evidence and the assumptions of the molecular clock model.

Phylogenies inferred from 'core' genes involved in translation and other conserved cellular processes have provided one of the few available windows into the earliest period of archaeal and bacterial evolution. However, core genes comprise only a small proportion of prokaryotic genomes and have sometimes been viewed as outliers (*Zhu et al., 2019*) in the sense that they are unusually vertical among prokaryotic gene families. This means that they are among the few prokaryotic gene families amenable to concatenation methods, which are useful for pooling signal from individual weakly resolved gene trees but which make the assumption that all sites evolve on the same underlying tree. If other gene families are included in concatenations, the results can be difficult to predict because differences in topology across sites are not modeled. Our analyses of the 381-gene expanded set suggest that this incongruence can lead to underestimation of the evolutionary distance between Archaea and Bacteria, in the sense of the branch length separating the archaeal and bacterial domains. We note that alternative conceptions of evolutionary distance are possible; for example, in a phenetic sense of overall genome similarity, extensive HGT will increase the evolutionary proximity (*Zhu et al., 2019*) of the domains so that Archaea and Bacteria may become intermixed at the single-gene level. While such data can encode an important evolutionary signal, it is not amenable to concatenation analysis. At the same time, it is clearly unsatisfactory to base our view of early evolution on a relatively small set of genes that appear to experience selective pressures rather distinct from the forces at play more broadly in prokaryotic genome evolution. These limitations are particularly unfortunate given the wealth of genome data now available to test hypotheses about early evolution. Exploring the evolutionary signal in more of the genome than hitherto is clearly a worthwhile endeavor. New methods, including more realistic models of gene duplication, transfer and loss (*Morel et al., 2021*; *Szöllősi et al., 2013*), and extensions to supertree methods to model paralogy (*Zhang et al., 2020*) and gene transfer, promise to enable genome-wide inference of prokaryotic history and evolutionary processes using methods that can account for the varying evolutionary histories of individual gene families.

# Materials and methods
## Data
We downloaded the individual alignments from *Zhu et al., 2019* (https://github.com/biocore/wol/tree/master/data/, *Zhu, 2022*), along with the genome metadata and the individual Newick files. We checked each published tree for domain monophyly and also performed AU (*Shimodaira, 2002*) tests to assess support for domain monophyly on the underlying sequence alignments using IQ-TREE 2.0.6 (*Minh et al., 2020*). The phylogenetic analyses were carried out using the 'reduced' subset of 1000 taxa outlined by the authors (*Zhu et al., 2019*) for computational tractability. These markers were trimmed according to the protocol in the original paper (*Zhu et al., 2019*), that is, sites with >90% gaps were removed, followed by removal of sequences with >66% gaps.

We also downloaded the *Williams et al., 2020* ('core'), *Petitjean et al., 2014* ('non-ribosomal'), and *Coleman et al., 2021* ('bacterial') datasets from their original publications.

## Annotations
Proteins used for phylogenetic analyses by *Zhu et al., 2019* were annotated to investigate the selection of sequences comprising each of the marker gene families. To this end, we downloaded the protein sequences provided by the authors from the following repository: https://github.com/biocore/wol/tree/master/data/alignments/genes. To obtain reliable annotations, we analyzed all sequences per

gene family using several published databases, including the arCOGs (version from 2014) (*Seemann, 2014*), KOs from the KEGG Automatic Annotation Server (KAAS; downloaded April 2019) (*Aramaki et al., 2020*), the PFAM database (release 31.0) (*Bateman et al., 2004*), the TIGRFAM database (release 15.0) (*Haft et al., 2003*), the Carbohydrate-Active enZymes (CAZy) database (downloaded from dbCAN2 in September 2019) (*Cantarel et al., 2009*), the MEROPs database (release 12.0) (*Rawlings et al., 2016*; *Saier et al., 2006*), the hydrogenase database (HydDB; downloaded in November 2018) (*Sondergaard et al., 2016*), the NCBI_ non-redundant (nr) database (downloaded in November 2018), and the NCBI COGs database (version from 2020). Additionally, all proteins were scanned for protein domains using InterProScan (v5.29–68.0; settings: `--iprlookup --goterms`) (*Jones et al., 2014*).

Individual database searches were conducted as follows: arCOGs were assigned using PSI-BLAST v2.7.1+ (settings: -evalue 1e-4 -show_gis -outfmt 6 -max_target_seqs 1000 -dbsize 100000000 -comp_based_stats F -seg no) (*Altschul et al., 1997*). KOs (settings: -E 1e-5), PFAMs (settings: -E 1e-10), TIGRFAMs (settings: -E 1e-20), and CAZymes (settings: -E 1e-20) were identified in all archaeal genomes using hmmsearch v3.1b2 (*Finn et al., 2011*). The MEROPs and HydDB databases were searched using BLASTp v2.7.1 (settings: -outfmt 6, -evalue 1e-20). Protein sequences were searched against the NCBI_nr database using DIAMOND v0.9.22.123 (settings: –more-sensitive –e-value 1e-5 –seq 100 –no-self-hits –taxonmap prot.accession2taxid.gz) (*Buchfink et al., 2015*). For all database searches, the best hit for each protein was selected based on the highest e-value and bitscore and all results are summarized in *Supplementary file 1* and full results are given in the Data Supplement: Expanded_Bacterial_Core_Nonribosomal_analyses/ Annotation_Tables/0_Annotation_ tables_full/All_Zhu_marker_annotations_16-12-2020.tsv.zip. For InterProScan, we report multiple hits corresponding to the individual domains of a protein using a custom script (parse_IPRdomains_ vs2_GO_2.py).

Assigned sequence annotations were summarized, and all distinct KOs and PFAMs were collected and counted for each marker gene. KOs and PFAMs with their corresponding descriptions were mapped to the marker gene file downloaded from the repository here and used in summarization of the 381 marker gene protein trees (*Supplementary file 1*).

For manual inspection of single marker gene trees, KO and PFAM annotations were mapped to the tips of the published marker protein trees, downloaded from the repository here. Briefly, the Genome ID, PFAM, PFAM description, KO, KO description, and NCBI Taxonomy string were collected from each marker gene annotation table and were used to generate mapping files unique to each marker gene phylogeny, which links the Genome ID to the annotation information (GenomeID|Domain|Pfam|Pfam Description|KO|KO Description). An in-house Perl script replace_tree_names.pl ( available here; *Dombrowski, 2022* copy archived at swh:1:rev:59ce418ec42160a15e82610120220b611b6e 96db) was used to append the summarized protein annotations to the corresponding tips in each marker gene tree. Annotated marker gene phylogenies were manually inspected using the following criteria, including (1) retention of reciprocal domain monophyly (Archaea and Bacteria) and (2) for the presence or absence of potential paralogous families. Paralogous groups and misannotated families present in the gene trees were highlighted and violations of search criteria were recorded in *Supplementary file 1*.

## Phylogenetic analyses

### COG assignment for the core, non-ribosomal, and bacterial marker genes

First, all gene sequences in the three published marker sets (core, non-ribosomal, and bacterial) were annotated using the NCBI COGs database (version from 2020). Sequences were assigned a COG family using hmmsearch v3.3.2 (*Finn et al., 2011*) (settings: -E 1e-5) and the best hit for each protein sequence was selected based on the highest e-value and bit score. To assign the appropriate COG family for each marker gene, we quantified the percentage distribution of all unique COGs per gene and selected the family representing the majority of sequences in each marker gene.

Accounting for overlap, this resulted in 95 unique COG families from the original 119 total marker genes across all three published datasets (*Supplementary file 2*). Orthologs corresponding to these 95 COG families were identified in the 700 genomes (350 Archaea, 350 Bacteria, *Supplementary file 3*) using hmmsearch v3.3.2 (settings: -E 1e-5). The reported BinID and protein accession were used to extract the sequences from the 700 genomes, which were used for subsequent phylogenetic analyses.

## Marker gene inspection and analysis

We aligned these 95 marker gene sequence sets using MAFFT-L-INS-i 7.475 (*Katoh and Toh, 2008*) and removed poorly aligned positions with BMGE 1.12 (*Criscuolo and Gribaldo, 2010*). We inferred initial ML trees (LG + G4 + F) for all 95 markers and mapped the KO and PFAM domains and descriptions, inferred from annotation of the 700 genomes, to the corresponding tips (see above). Manual inspection took into consideration monophyly of Archaea and Bacteria and the presence of paralogs, and other signs of contamination (HGT, LBA). Accordingly, single-gene trees that failed to meet reciprocal domain monophyly were excluded, and any instances of HGT, paralogous sequences, and LBA artifacts were manually removed from the remaining trees, resulting in 54 markers across the three published datasets that were subject to subsequent phylogenetic analysis (LG + C20 + G4 + F) and further refinement (see below).

## Ranking markers based on split score

We applied an automated marker gene ranking procedure devised previously (the split score, *Dombrowski et al., 2020*) to rank each of the 54 markers that satisfied reciprocal monophyly based on the extent to which they recovered established phylum-, class-, or order-level relationships within the archaeal and bacterial domains (*Supplementary file 4*).

The script quantifies the number of splits, or occurrences where a taxon fails to cluster within its expected taxonomic lineage, across all gene phylogenies. Briefly, we assessed monophyletic clustering using phylum-, class-, and order-level clades within Archaea (Cluster1) in combination with Cluster0 (phylum) or Cluster3 (i.e., on class-level if defined and otherwise on phylum-level; *Supplementary file 4*) for Bacteria. We then ranked the marker genes using the following split score criteria: the number of splits per taxon and the splits normalized to the species count. The percentage of split phylogenetic groups was used to determine the highest ranking (top 50%) markers.

## Concatenation

Based on the split score ranking of the 54 marker genes (above), the top 50% (27 markers, *Supplementary file 4*) marker genes were manually inspected using criteria as defined above, and contaminating sequences were manually removed from the individual sequence files. Following inspection, marker protein sequences were aligned using MAFFT-L-INS-i 7.475 (*Katoh and Standley, 2013*) and trimmed using BMGE (version 1.12, under default settings) (*Criscuolo and Gribaldo, 2010*). We concatenated the 27 markers into a supermatrix, which was used to infer an ML tree (*Figure 4*, under LG + C60 + G4 + F), evolutionary rates (see below), and rate category supermatrices, as well as to perform model performance tests (see below). We also concatenated the non-ribosomal and ribosomal markers from the 27 and 54 marker sets into four more supermatrices and inferred ML trees under (LG + C60 + G4 + F) (*Table 1*). Two additional supermatrices were constructed from the 54 markers, one before manual removal of apparent HGTs and one after the removal, with both sets of markers aligned and trimmed in the same way as the other datasets (see above). We also inferred an ML tree under LG + C60+ G4 + F from a supermatrix consisting of a concatenation of 25 marker genes after removing COG0480 and COG5257.

## Constraint analysis

We performed an ML free topology search using IQ-TREE 2.0.6 (*Minh et al., 2020*) under the LG + G4 + F model, with 1000 ultrafast bootstrap replicates (*Hoang et al., 2018*) on each of the markers from the expanded, bacterial, core, and non-ribosomal sets. We also performed a constrained analysis with the same model in order to find the ML tree in which Archaea and Bacteria were reciprocally monophyletic. For the expanded set, we plotted branch lengths from the maximum likelihood trees constrained to recover domain monophyly; for the other datasets, we plotted branch lengths from the maximum likelihood trees. We then compared both trees using the AU (*Shimodaira, 2002*) test in IQ-TREE 2.0.6 (*Minh et al., 2020*) with 10,000 RELL (*Shimodaira, 2002*) bootstrap replicates. To evaluate the relationship between marker gene verticality and AB branch length, we calculated the difference in log-likelihood between the constrained and unconstrained trees in order to rank the genes from the expanded marker set. We then concatenated the top 20 markers (with the lowest difference in log-likelihood between the constrained and unconstrained trees) and iteratively added five markers with the next smallest difference in log-likelihood to the concatenate; this was repeated

until we had concatenates up to 100 markers (with the lowest difference in log-likelihood) we inferred trees under LG + C10 + G4 + F in IQ-TREE 2.0.6, with 1000 ultrafast bootstrap replicates and calculated AB length.

### Site and gene evolutionary rates

We inferred rates using the `--rate` option in IQ-TREE 2.0.6 (*Minh et al., 2020*) for both the 381 marker concatenation from *Zhu et al., 2019* and the top 5% of marker genes based on the results of difference in log-likelihood between the constrained tree and free-tree search in the constraint analysis (above). We also used this method to explore the differences in rates for the 27 marker set. We built concatenates for sites in the slowest and fastest rate categories, and inferred branch lengths from each of these concatenates using the tree inferred from the corresponding dataset as a fixed topology.

### Substitution model fit

Model fit tests were undertaken using the top 5% concatenate described above, with the alignment being trimmed with BMGE 1.12 (*Criscuolo and Gribaldo, 2010*) with default settings (BLOSUM62, entropy 0.5) for all of the analyses except the 'untrimmed' LG + G4 + F run; other models on the trimmed alignment were LG + G4 + F, LG + R4 + F and LG + C10,20,30,40,50,60 + G4 + F, with 1000 ultrafast (*Hoang et al., 2018*) bootstrap replicates. Model fitting was done using ModelFinder (*Kalyaanamoorthy et al., 2017*) in IQ-TREE 2.0.6 (*Minh et al., 2020*). For the 27 marker concatenation, we performed a model finder analysis (-m MFP) including additional complex models of evolution (i.e., LG + C60 + G4 + F, LG + C50 + G4 + F, LG + C40 + G4 + F, LG + C30 + G4 + F, LG + C20 + G4 + F, LG + C10 + G4 + F, LG + G4 + F, LG + R4+ F) to the default, to find the best-fitting model for the analysis. This revealed that, according to AIC (Akaike information criterion), BIC (Bayesian information criterion), and cAIC (corrected Akaike information criterion), LG + C60 + G4 + F was the best-fitting model. For comparison, we also performed analyses using the following models: LG + G4 + F, LG + C20 + G4 + F, LG + C40 + G4 + F (*Table 1*).

### Molecular clock analyses

Molecular clock analyses were devised to test the effect of genetic distance on the inferred age of LUCA. Following the approach of *Zhu et al., 2019*, we subsampled the alignment to 100 species. Five alternative alignments were analyzed, representing conserved sites across the entire alignment, randomly selected sites across the entire alignment, only ribosomal marker genes, the top 5% of marker genes according to ΔLL, and the top 5% of marker genes further trimmed under default settings in BMGE 1.12 (*Criscuolo and Gribaldo, 2010*). Divergence time analyses were performed in MCMCTree (*Yang, 2007*) under a strict clock model. We used the normal approximation approach, with branch lengths estimated in codeml under the LG + G4 model. In each case, a fixed tree topology was used alongside a single calibration on the Cyanobacteria-Melainabacteria split. The calibration was modeled as a uniform prior distribution between 2.5 and 2.6 Ga, with a 2.5% probability that either bound could be exceeded. For each alignment, four independent MCMC chains were run for 2,000,000 generations to achieve convergence.

We repeated clock analyses under a relaxed (independent rates drawn from a lognormal distribution) clock model with an expanded sampling of fossil calibration (*Supplementary file 5b*). We repeated the analyses with two approaches to define the maximum age calibration. The first used the moon-forming impact (4.52 Ga) under the provision that no forms of life are likely to have survived this event. The second relaxed this assumption, instead using the estimated age of the universe (13.7 Ga) as a maximum. Analyses were performed as above.

### Split score analysis for expanded set markers

We used the previously described split score ranking procedure to quantify the number of taxonomic splits in the 381 marker gene phylogenies generated using the 1000-taxa subsample defined by *Zhu et al., 2019*. Taxonomic clusters were assigned using the Genome Taxonomy Database (GTDB) taxonomic ranks downloaded from the repository (https://github.com/biocore/wol/tree/master/data/taxonomy/gtdb). Lineage-level monophyly was defined at the class level for all archaea (Arc1) and the phylum level for all bacteria (Bac0) (*Supplementary file 1*).

Of the original 10,575 genomes, 843 lacked corresponding GTDB assignments. For complete taxonomic coverage of the dataset, we used the GTDB Toolkit (GTDB-Tk) v0.3.2 (*Chaumeil et al., 2019*) to classify these genomes based on GTDB release 202. One of the 843 unclassified taxa (gid: G000715975) failed the GTDB-Tk quality control check, resulting in no assignment; therefore, we manually assigned this taxon to the Actinobacteriota based on the corresponding affiliation to the Actinobacteria in the NCBI taxonomic ranks provided in the genomic metadata downloaded from the repository (https://github.com/biocore/wol/tree/master/data/genomes). Additionally, two archaeal taxa within the Poseidoniia_A (gids: G001629155, G001629165) were manually assigned to the archaeal class MGII (*Supplementary file 1*).

## Plotting

Split score statistical analyses were performed using R 3.6.3 (R Core Team, 2020). All other statistical analyses were performed using R 4.0.4 (*R Development Core Team, 2021*), and data were plotted with ggplot2 (*Wickham, 2009*).

## Acknowledgements

This work was supported by the Gordon and Betty Moore Foundation through grant GBMF9741 to TAW, AS, and GJSz. ERRM was supported by a Royal Society Enhancement Award (RGF\EA\180199) to TAW. CP was supported by NERC grant NE/P00251X/1 to TAW. TAW was supported by a Royal Society University Research Fellowship (URF\R\201024). GJSz received funding from the European Research Council under the European Union's Horizon 2020 research and innovation program under Grant Agreement 714774 and Grant GINOP-2.3.2.-15-2016-00057. AS received funding from the Swedish Research Council (VR starting grant 2016-03559), the NWO-I foundation of the Netherlands Organisation for Scientific Research (WISE fellowship), and the European Research Council (ERC Starting grant 947317, ASymbEL). ND was supported through the WISE fellowship, ERC StG 947317 and GBMF9741 to AS.

## Additional information

### Funding

| Funder | Grant reference number | Author |
|---|---|---|
| Gordon and Betty Moore Foundation | GBMF9741 | Anja Spang<br>Tom A Williams |
| Royal Society | RGF\EA\180199 | Edmund RR Moody<br>Tom A Williams |
| Natural Environment Research Council | NE/P00251X/1 | Celine Petitjean<br>Tom A Williams |
| Royal Society | URF\R\201024 | Tom A Williams |
| H2020 European Research Council | 714774 | Gergely J Szöllősi |
| H2020 European Research Council | GINOP-2.3.2.-15-2016-00057 | Gergely J Szöllősi |
| Swedish Research Council | 2016-03559 | Anja Spang |
| Netherlands Organisation for Scientific Research | WISE Fellowship | Anja Spang |
| H2020 European Research Council | 947317 | Anja Spang |

The funders had no role in study design, data collection and interpretation, or the decision to submit the work for publication.

## Author contributions
Edmund RR Moody, Tara A Mahendrarajah, Nina Dombrowski, Conceptualization, Data curation, Formal analysis, Investigation, Methodology, Writing – original draft, Writing – review and editing; James W Clark, Data curation, Formal analysis, Investigation, Methodology, Writing – original draft, Writing – review and editing; Celine Petitjean, Data curation, Formal analysis, Writing – review and editing; Pierre Offre, Conceptualization, Data curation, Formal analysis, Writing – review and editing; Gergely J Szöllősi, Conceptualization, Formal analysis, Investigation, Methodology, Writing – original draft, Writing – review and editing; Anja Spang, Conceptualization, Data curation, Formal analysis, Funding acquisition, Investigation, Methodology, Supervision, Writing – original draft, Writing – review and editing; Tom A Williams, Conceptualization, Formal analysis, Funding acquisition, Investigation, Methodology, Supervision, Writing – original draft, Writing – review and editing

## Author ORCIDs
Edmund RR Moody ⓘ http://orcid.org/0000-0002-8785-5006
Tara A Mahendrarajah ⓘ http://orcid.org/0000-0001-7032-6581
Nina Dombrowski ⓘ http://orcid.org/0000-0003-1917-2577
Pierre Offre ⓘ http://orcid.org/0000-0002-3660-2164
Gergely J Szöllősi ⓘ http://orcid.org/0000-0002-8556-845X
Anja Spang ⓘ http://orcid.org/0000-0002-6518-8556
Tom A Williams ⓘ http://orcid.org/0000-0003-1072-0223

## Decision letter and Author response
Decision letter https://doi.org/10.7554/eLife.66695.sa1
Author response https://doi.org/10.7554/eLife.66695.sa2

# Additional files

## Supplementary files
• Supplementary file 1. Marker metadata, KO and Pfam annotations and descriptions, and manual inspection notes for reciprocal monophyly and presence of paralogs for 381 marker genes used in *Zhu et al., 2019* (Materials and methods).

• Supplementary file 2. Marker metadata, KO and Pfam annotations and descriptions, and manual inspection notes for 95 markers in the core, bacterial, and non-ribosomal marker gene sets (Materials and methods).

• Supplementary file 3. NCBI taxonomic information for 350 archaeal and 350 bacterial genomes sampled in the new analyses.

• Supplementary file 4. Clade definitions for quantifying taxonomic splits and split score statistical summaries for ranking of the core, bacterial, non-ribosomal marker genes, and 381 marker genes (Materials and methods).

• Supplementary file 5. Annotations of the top 20 genes from the expanded set, and a list of fossil calibrations. (a) Functional annotations for the top 20 genes used and in *Figure 6* and referred to in *Figure 6A*. (b) A list of fossil calibrations employed in relaxed molecular clock analyses. All calibrations were modeled as uniform distributions between a hard minimum and a soft maximum. The probability that the maximum could be exceeded was modeled as a 2.5% probability tail.

• Transparent reporting form

## Data availability
All of the data, including sequence alignments, trees, annotation files, and scripts associated with this manuscript have been deposited in the FigShare repository at https://doi.org/10.6084/m9.figshare.13395470.

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
