## [Editor Report]

This contribution is of interest to molecular phylogeny scientists, in particular, and to a broad public interested in early evolution, in general, as it elegantly supports the long-standing (but recently challenged) hypothesis that bacteria and archaea are separated by a long branch.

---

## [Decision Letter]

**Decision letter after peer review:**

Thank you for submitting your article "An estimate of the deepest branches of the tree of life from ancient vertically-evolving genes" for consideration by *eLife*. Your article has been reviewed by 3 peer reviewers, and the evaluation has been overseen by George Perry as the Senior Editor. The following individual involved in review of your submission has agreed to reveal their identity: Eugene V Koonin (Reviewer #2).

The reviewers have discussed their reviews with one another, and the Senior Editor has drafted this to help you prepare a revised submission.

*Reviewer #1 (Recommendations for the authors):*

There are seven figures in this manuscript, but several of them are simple in both visual content and scientific information it delivers. Specifically, I think Figures 1-4 can be merged into one or two figures.

*Reviewer #2 (Recommendations for the authors):*

This is an excellent, carefully performed and very clearly presented work, so I do not have any truly major criticisms. I strongly believe that the long AB branch reflects reality.

However, as mentioned in the public remarks as well, I think more caution is due in the interpretation of the phylogenies inside archaea and bacteria, especially because this was done on the same 27 gene set that was used for the universal tree whereas more markers are certainly available for archaea and bacteria taken separately. To me, the results of these analysis are somewhat suspicious in that the archaeal superphylum that involves (putative) parasites/symbionts with tiny genomes (DPANN) remains basal whereas CPR moves into the midst of the bacterial tree. Is there a chance that the deep position of DPANN is also due to LBA? In any case, I think this issues needs to be discussed in the paper, and some notes of caution should be included.

[Editors’ note: further revisions were suggested prior to acceptance, as described below.]

Thank you for submitting your revised article "An estimate of the deepest branches of the tree of life from ancient vertically-evolving genes" for consideration by *eLife*. Your article has been reviewed by 3 peer reviewers, and the evaluation has been overseen by George Perry as the Senior Editor.

There is remaining disparity among the reviewers concerning your revised paper. Reviewers 2 and 3 are approving of your revision and the manuscript overall; Reviewer 1 remains more critical.

I plan to accept your manuscript once you compose a response to the below comments from Reviewer 1 alongside any revisions you choose to make, seeing the disagreeing reviews and your responses that will be published alongside the paper as a healthy part of the process and the scientific record.

*Reviewer #1:*

Overall, this revision has improved from the previous version. I am glad that the authors added tests on additional metrics, including relative AB distance and split scores, as suggested by reviewer #3 and me. I am also glad with the additional discussion on early evolution, in response to reviewer #2 and me. The discussion on the definition of "evolutionary distance" and the refinement of the scope on "branch length" are reasonable, leaving this an open question for researchers and philosophers. The authors admitted that core genes are outliers (line 537 and in the response letter) of the prokaryotic genome, which I appreciate. Meanwhile, the authors have provided more robust evidence showing that ribosomal proteins are not evolving faster than other vertically evolving genes. This result is informative and valuable. The new guideline of iterative manual curation of marker gene families provided in Figure 6 is very helpful to the audience. The authors have also done good job in resolving some of my other concerns

However, I do think that several of my major concerns have not been sufficiently addressed, leaving me unconvinced of the main conclusion: AB branch length is naturally long. Actually, my concern grew stronger after seeing the new Figure 5B, which itself is an excellent analysis. My opinions are detailed below, with section titles corresponding to the titles in my original comments.

"## Verticality is not causative of short AB branch"

In this section I pointed out a logical issue in the current study. The authors agreed. Instead of the analysis I suggested (which is non-trivial, though), the authors performed a different analysis to address this issue: rooted gene trees indicate a slower evolutionary rate in more vertical genes. I appreciate this additional analysis. However, I would like to point out that it does not directly solve the issue. My original concern was that "the most vertical genes happen to be those that evolved faster *at the AB split*." What the authors tested was the *overall* evolutionary rate of each gene family. It is expected that core genes are more conserved in general because they bear important functions. See also my comment below.

"## AB branch by ribosomal and non-ribosomal genes" and a few other related comments

The authors extended their discussion on the accelerated ribosomal protein evolution hypothesis suggested by Petitjean et al., 2014 and Zhu et al., 2019. I appreciate this discussion, which adds valuable information and opinion to this topic.

There is one logic issue though: the phrase "accelerated evolution of ribosomal proteins", to my understanding, means that "they evolved faster during AB split *than themselves in later time*". However, the hypothesis being examined in this work was "they evolved faster *than other core genes* during AB split", by showing that ribosomal proteins do not indicate a longer AB branch length compared with vertically evolving, non-ribosomal genes. The first point was not rebutted. Instead (to my curiosity), it was supported by the new Figure 5B, which reveals a much shorter AB branch after time calibration.

A second component of my comment, which was not addressed (I apologize if I missed it), is that even within the set of 54 vertically evolving genes analyzed in this work, there are outliers. My original comment stated: "A closer look at Figures 3 and 4 suggests that while the AB lengths indicated by the ribosomal markers are within a relatively narrow range, those by the non-ribosomal ones are very diverse, including ones that are several folds longer than the ribosomal average." "…the figures clearly show that there are outliers even in this small set". In the revised manuscript, the authors have re-drawn Figure 4 (now Figure 2B) with its y-axis log-transformed. But this does not solve the problem. This pattern leaves a question to the reader: Given that the AB branch lengths estimated by different core genes are so diverse, which ones are better indicators of the true AB branch length than others?

"## AB distance metric"

I appreciate that the authors included the relative AB distance in the comparative analysis. The revised Figure 1 is notably more informative to the audience. I have one suggestion: the authors may label the test statistics (R and p-value) in the figure panels so that one can immediately see how strong the correlations are. A regression line with confidence interval alone may be less informative (especially under log scale) with statistics provided elsewhere.

For example, it is evident that AB branch length and relative AB distance are strongly positively correlated (R = 0.74) (panel B). However, the negative correlations indicated in panel C-F are weak (R = -0.22 to -0.30), and weaker than a similar statistic reported in Zhu et al.,’ Figure 5e: R2 = 0.157 (that is, R = 0.40), although the latter was investigating overall topological incongruence instead of AB separation. The authors should at least mention Zhu et al.,' result in the text.

I also appreciate that the authors acknowledge the various definitions of evolutionary distance, and revised the manuscript accordingly.

"## Concatenation and branch length"

In this comment I proposed that "The best answer is to conduct an analysis to show that concatenating genes with conflicting phylogeny would result in an AB branch that is *shorter than the mean* of those genes, and the reduction of AB branch length is correlated with the amount of conflict involved."

The authors addressed my comments with three arguments: First, they pointed out that many of the expanded marker genes are likely not ancient and don't contribute to AB branch length. Second, they found that "as more markers were added (with lower verticality), the inferred AB branch length from the concatenate was reduced." Third, they took marker genes and "concatenated them before and after pruning of unambiguous HGTs", and found that the latter indicated a longer AB branch (but the result is not presented in the manuscript; I apologize if I missed it).

These analyses are valuable. However, I am afraid that none of the analyses directly address my comment. To clarify: in order to effectively rebut the concatenation method, one needs to demonstrate that *it is the act of concatenation*, instead of including genes with lower verticality, that reduces the AB branch length. I understand that it may not be trivial. But since the authors stated that "this (concatenation followed by branch length estimation) is our main critique of the Zhu et al., analyses" (which I agree to certain extent; see my previous comment), robust and compelling analyses are needed to strengthen the point, otherwise it should not be emphasized.

"## Divergence time estimation"

I am glad to see that the authors have significantly expanded the divergence time estimation in response to my and reviewer #3's comments. In particular, Figure 5B is very useful, because it provides a map of inferred evolutionary rate over time and lineages. The authors did a good job in discussing the result in a more neutral and constructive way (DTE is difficult), especially in explaining the outlier in that figure panel (LACA). I agree with the argument in its current form, that the lack of maximum age calibration causes difficulties.

However, I would like to note two important observations from Figure 5B: First, the AB branch length revealed by this time-calibrated tree is much shorter than that by the non-time-calibrated trees (e.g., Figures 3 and 6). This pattern is not a good echo of the main point defended in this manuscript (AB branch is long). I wouldn't assert that they contradict, since the branch lengths in the two trees are of different units (geological time vs substitutions per site). However, divergence time is another, and more intuitive definition of "evolutionary distance". In that sense, the short AB distance is not supported by this result.

Second, the result shows that LACA is an outlier in terms of evolutionary rate. This pattern actually supports Petitjean et al., and Zhu et al.,'s hypothesis that there was accelerated evolution during the AB split. Although the authors discussed this result and attributed it to the maximum age calibration, but they did not provide the evolutionary rate estimated without this calibration (LBCA > 3.227Ga). I suspect that in any circumstance, either or both of LACA and LBCA branches would have much higher evolutionary rate than the later diversification of the two domains.

Taken together, I think the DTE result presented in this manuscript is quite inline with Zhu et al., despite the different explanations. Therefore I remain reserved on the strength of the authors' argument.

---

## [Author Response]

Reviewer #1 (Recommendations for the authors):There are seven figures in this manuscript, but several of them are simple in both visual content and scientific information it delivers. Specifically, I think Figures 1-4 can be merged into one or two figures.

Thanks for this suggestion. In the revised manuscript, both Figure 1 and the former Figure 2 (now Figure 5) have been updated to include substantial additional analyses. For the former Figures 3-4, we agree that they can be merged, and have now done so.

Reviewer #2 (Recommendations for the authors):This is an excellent, carefully performed and very clearly presented work, so I do not have any truly major criticisms. I strongly believe that the long AB branch reflects reality.However, as mentioned in the public remarks as well, I think more caution is due in the interpretation of the phylogenies inside archaea and bacteria, especially because this was done on the same 27 gene set that was used for the universal tree whereas more markers are certainly available for archaea and bacteria taken separately. To me, the results of these analysis are somewhat suspicious in that the archaeal superphylum that involves (putative) parasites/symbionts with tiny genomes (DPANN) remains basal whereas CPR moves into the midst of the bacterial tree. Is there a chance that the deep position of DPANN is also due to LBA? In any case, I think this issues needs to be discussed in the paper, and some notes of caution should be included.

Deep phylogeny is uncertain, current models are far from perfect, and so it is indeed difficult to exclude the possibility that the basal position of DPANN is due to LBA. We have now discussed these difficulties when presenting our estimate of the universal tree, writing:

“A basal placement of DPANN within Archaea is sometimes viewed with suspicion (Aouad et al., 2018) because DPANN genomes are reduced and appear to be fast-evolving, properties that may cause LBA artifacts (Dombrowski et al., 2019)when analyses include Bacteria. However, in contrast to CPR, with which DPANN share certain ecological and genomic similarities (e.g. host dependency, small genomes, limited metabolic potential), the early divergence of DPANN from the archaeal branch has received support from a number of recent studies (Williams, 2017; Dombrowski, 2020; Rinke, 2021) though the inclusion of certain lineages within this radiation remains controversial (Aouad et al., 2018; Feng et al., 2021). While more in-depth analyses will be needed to further illuminate the evolutionary history of DPANN and establish which archaeal clades constitute this lineage, our work is in agreement with current literature and a recently established phylogeny-informed archaeal taxonomy (Rinke et al., 2021).”

[Editors’ note: further revisions were suggested prior to acceptance, as described below.]

Reviewer #1:Overall, this revision has improved from the previous version. I am glad that the authors added tests on additional metrics, including relative AB distance and split scores, as suggested by reviewer #3 and me. I am also glad with the additional discussion on early evolution, in response to reviewer #2 and me. The discussion on the definition of "evolutionary distance" and the refinement of the scope on "branch length" are reasonable, leaving this an open question for researchers and philosophers. The authors admitted that core genes are outliers (line 537 and in the response letter) of the prokaryotic genome, which I appreciate. Meanwhile, the authors have provided more robust evidence showing that ribosomal proteins are not evolving faster than other vertically evolving genes. This result is informative and valuable. The new guideline of iterative manual curation of marker gene families provided in Figure 6 is very helpful to the audience. The authors have also done good job in resolving some of my other concernsHowever, I do think that several of my major concerns have not been sufficiently addressed, leaving me unconvinced of the main conclusion: AB branch length is naturally long. Actually, my concern grew stronger after seeing the new Figure 5B, which itself is an excellent analysis. My opinions are detailed below, with section titles corresponding to the titles in my original comments.

We thank the reviewer for responding constructively to our revision and previous comments. While we address their new comments in detail below, it is perhaps worth beginning by setting out what we see as the underlying disagreement with this reviewer, which centres on what “core” genes are, how they evolved, why their AB branch lengths are longer than those of other genes that are encoded on bacterial and archaeal genomes. The divergence of views might be summarised in the following way:

One view begins by considering extant prokaryotic gene families broadly. It notes that the “core” genes that have traditionally been used to infer trees of life make up only a small fraction of prokaryotic genetic diversity, and are outliers in a number of ways, including AB branch length. On this view, a more representative estimate of the AB branch length can be obtained by considering all, or a reasonable subset, of the genes that are encoded on modern bacterial and archaeal genomes. We take this to be the rationale for Zhu et al.,’s concatenation analysis of the expanded gene set to estimate the AB branch length in substitutions per site. The resulting estimate was shorter than the AB length inferred solely from core genes, motivating the view that previous analyses have been misled by an overly restrictive focus on core genes for estimating deep phylogeny.

We disagree with this view, although we are sympathetic to the ultimate goal of incorporating as much of the data as possible to interrogate deep phylogeny. We note that AB branch lengths from different genes can only be combined using concatenation if the genes share a common evolutionary history of vertical descent. This implies that marker genes for resolving the universal tree must be both ancient and vertically-evolving: that is, they should have been already present in LUCA (so that they can inform on LUCA-LACA and LUCA-LBCA evolution) and should have evolved vertically since. When we select genes that best fit these criteria, the resulting set has long AB branches, leading to inference of a long AB branch length (measured in substitutions per site). Other gene families that are found in both Archaea and Bacteria, but that do not meet the criteria for inclusion in concatenation analysis are less likely to have evolved vertically and suggest a shorter AB branch length. However, this apparently shorter branch length is artifactual, resulting from the inclusion of gene families that have evolved post-LBCA/LACA, or alternatively, that have experienced extensive HGT in the time since LUCA. Our analyses have demonstrated that adding such genes to concatenates reduces the inferred AB branch length due to phylogenetic discord between the constituent marker genes.

Finally, an additional strand of this interesting debate relates to the possibility of accelerated evolution of ribosomal genes, but not other kinds of ancient genes, during the divergence of Archaea and Bacteria. Since the set of ancient, vertically-evolving genes is enriched for ribosomal proteins, might an acceleration of this kind lead to overestimation of the AB branch length? Our analyses do not provide support for this hypothesis because we find no significant difference between AB branch lengths of ribosomal and non-ribosomal proteins in the ancient, vertically-evolving marker gene set (p = 0.6191, Wilcoxon rank-sum test). However, when taken together with geological evidence for crown-group Bacteria being older than ~3.225Ga (Betts et al., 2018), the long AB branches inferred from vertically-evolving genes is consistent with higher *genome-wide* rates of molecular evolution during the early history of life (by “genome-wide”, we mean higher evolutionary rates, i.e., substitutions/site/year, across all vertically-evolving genes from that period that have left descendants in sampled modern taxa). This inference, however, is tentative because the paucity of available data and calibrations result in uncertain estimates for rates and times on the deepest branches of the tree of life.

Responses to the reviewer’s specific comments follow.

"## Verticality is not causative of short AB branch"In this section I pointed out a logical issue in the current study. The authors agreed. Instead of the analysis I suggested (which is non-trivial, though), the authors performed a different analysis to address this issue: rooted gene trees indicate a slower evolutionary rate in more vertical genes. I appreciate this additional analysis. However, I would like to point out that it does not directly solve the issue. My original concern was that "the most vertical genes happen to be those that evolved faster at the AB split." What the authors tested was the overall evolutionary rate of each gene family. It is expected that core genes are more conserved in general because they bear important functions. See also my comment below.

Our analyses demonstrate that vertically-evolving genes (whether ribosomal or non-ribosomal, and whether verticality is measured between or within domains) have longer AB stems, and that these longer stems are not due to a higher overall evolutionary rate in high-verticality genes. The reviewer suggests the possibility that these long AB branches might be due to a rate shift that only occurred in vertically-evolving genes: they evolved fast during the divergence of Archaea and Bacteria but more slowly afterwards, resulting in long AB stems but a lower overall evolutionary rate compared to less vertical genes. As we note above, and as we wrote in the previous version of our manuscript (“calibrations within Bacteria imply that LBCA cannot be younger than 3.227 Ga…as a result, with a 4.52Ga maximum the LUCA to LBCA branch cannot be longer than 1.28Ga.”), our results do not exclude the possibility of a faster genome-wide evolutionary rate during early evolution.

Where we differ with R1 is on the question of whether the most vertically-evolving genes might have experienced additional rate acceleration as a category compared to other genes present during that period. Our analyses suggest that, beyond the set of vertically-evolving genes identified in a range of previous studies (and the consensus set assembled here), it is difficult to be sure that other genes included in the expanded marker set map back to LUCA; in cases where they may do, the complex post-LUCA evolutionary histories of the genes make curation and inclusion in concatenations problematic. Manual inspection of the lower-verticality genes of the expanded set revealed that 317 lack an unambiguous branch that can be mapped to the original divergence of Archaea and Bacteria, so cannot directly speak to the AB branch length; while 246 show mixing of Archaea and Bacteria from different paralogues (Supplementary File 1). When forced to follow the concatenated species tree, these genes support a short but artificial AB branch which is not present in their maximum likelihood gene trees. Thus, the different AB branch length distributions in vertically-evolving compared to less vertically-evolving genes reflect a difference in the age and evolutionary mode of the genes, rather than distinct rates of sequence evolution during a common shared history on the AB stem prior to the radiation of the archaeal and bacterial domains.

"## AB branch by ribosomal and non-ribosomal genes" and a few other related commentsThe authors extended their discussion on the accelerated ribosomal protein evolution hypothesis suggested by Petitjean et al., 2014 and Zhu et al., 2019. I appreciate this discussion, which adds valuable information and opinion to this topic.There is one logic issue though: the phrase "accelerated evolution of ribosomal proteins", to my understanding, means that "they evolved faster during AB split than themselves in later time". However, the hypothesis being examined in this work was "they evolved faster than other core genes during AB split", by showing that ribosomal proteins do not indicate a longer AB branch length compared with vertically evolving, non-ribosomal genes. The first point was not rebutted. Instead (to my curiosity), it was supported by the new Figure 5B, which reveals a much shorter AB branch after time calibration.

This exchange helps to clarify part of the disagreement here, as – in contrast to the reviewer – we take the hypothesis of accelerated ribosomal protein evolution to imply that ribosomal proteins evolved faster than other proteins along the AB branch. We note that the observation originally underlying the hypothesis was that a ribosomal concatenate had a longer absolute AB branch length than a non-ribosomal concatenate, suggesting accelerated ribosomal evolution on the AB branch (Petitjean et al., 2014). As noted above, our analyses suggest it is possible, though by no means certain, that the evolutionary rate of all genes was higher during early evolution, but we found no difference between ribosomal and non-ribosomal genes. To investigate the possibility of a greater post-stem rate shift in ribosomal compared to non-ribosomal genes, we have now also calculated the AB lengths, total tree lengths, and associated rates for the ribosomal and non-ribosomal subsets of the 27- and 54-gene concatenates (see the new Table 1). The values do not support accelerated evolution of ribosomal proteins on the AB stem either in absolute terms or in comparison to other ancient proteins, nor are they consistent with a more pronounced rate shift in ribosomal proteins following the radiation of the archaeal and bacterial domains: for example, the AB branch makes up ~0.7% of the total tree length of the ribosomal concatenate in both datasets, while the same branch comprises ~1.5% and 0.88% of tree length for the 27- and 54-gene non-ribosomal datasets, respectively. We have updated the manuscript to include these new analyses, and also to clarify what we mean by “accelerated evolution of ribosomal proteins”. We now write:

“To investigate further, we compared AB branch lengths inferred from concatenates of the ribosomal and non-ribosomal subsets of the 54 ancient, vertically-evolving genes (Table 1). AB branch lengths from the ribosomal and non-ribosomal concatenates were similar overall, with some support for a longer AB branch length from vertically-evolving non-ribosomal genes. Thus, these data do not support an accelerated evolutionary rate for ribosomal genes compared to other kinds of genes on the AB branch.”

A second component of my comment, which was not addressed (I apologize if I missed it), is that even within the set of 54 vertically evolving genes analyzed in this work, there are outliers. My original comment stated: “A closer look at Figures 3 and 4 suggests that while the AB lengths indicated by the ribosomal markers are within a relatively narrow range, those by the non-ribosomal ones are very diverse, including ones that are several folds longer than the ribosomal average.” “…the figures clearly show that there are outliers even in this small set”. In the revised manuscript, the authors have re-drawn Figure 4 (now Figure 2B) with its y-axis log-transformed. But this does not solve the problem. This pattern leaves a question to the reader: Given that the AB branch lengths estimated by different core genes are so diverse, which ones are better indicators of the true AB branch length than others?

The reviewer is correct that inferred AB branch lengths vary among the set of genes which, according to our analyses, map to LUCA and can therefore be used to infer the AB branch length, though they are clearly different to most of the expanded set (expanded set AB length mean: 0.1383, variance: 0.4484, 54-markers AB length mean: 1.5992, variance = 2.317, 27-markers AB length mean: 1.8486, variance: 1.42). In general, branch length variation in single gene trees is the norm, and likely reflects variation in functional constraints among genes and lineages, as well as the difficulties in mapping substitution histories on ancient branches of gene trees. Our answer to the reader would therefore be that the AB branch lengths of these genes are noisy estimates of an underlying genome-wide substitution rate averaged over the period in which Archaea and Bacteria diverged; the single gene branch lengths reflect this underlying rate but also gene-specific selective pressures, among-gene variation in coalescence time prior to the divergence of the archaeal and bacterial species-level lineages, stochastic noise and the limitations of the substitution model. Since these genes appear to share a common species phylogeny (within the limits of resolution of our analysis), a reasonable point estimate for the length of any branch might be the maximum likelihood branch length from concatenation (or a distribution of sampled branch lengths from an MCMC analysis), and we do not see a compelling reason to treat the AB branch differently to other species tree branches in this case. We note that none of this excludes the possibility that some of the gene-specific effects on rate may be biologically interesting in their own right, and our plot (Figure 2B) and the associated data (Supplementary File 4, Splitscore_Ablength_treelength) will allow interested readers to investigate particular outliers in greater detail.

“## AB distance metric”I appreciate that the authors included the relative AB distance in the comparative analysis. The revised Figure 1 is notably more informative to the audience. I have one suggestion: the authors may label the test statistics (R and p-value) in the figure panels so that one can immediately see how strong the correlations are. A regression line with confidence interval alone may be less informative (especially under log scale) with statistics provided elsewhere.For example, it is evident that AB branch length and relative AB distance are strongly positively correlated (R = 0.74) (panel B). However, the negative correlations indicated in panel C-F are weak (R = -0.22 to -0.30), and weaker than a similar statistic reported in Zhu et al.,' Figure 5e: R2 = 0.157 (that is, R = 0.40), although the latter was investigating overall topological incongruence instead of AB separation. The authors should at least mention Zhu et al.,' result in the text.I also appreciate that the authors acknowledge the various definitions of evolutionary distance, and revised the manuscript accordingly.

We have now labelled the test statistics as suggested, and have modified the text to note the agreement between our analyses and the quartet score/relative AB distance metric comparison in Figure 5E of Zhu et al., (2019), writing:

“Indeed, Zhu et al., (2019) also recovered a significant positive relationship between gene verticality and relative AB distance (see their Figure 5E).”

"## Concatenation and branch length"In this comment I proposed that "The best answer is to conduct an analysis to show that concatenating genes with conflicting phylogeny would result in an AB branch that is shorter than the mean of those genes, and the reduction of AB branch length is correlated with the amount of conflict involved."The authors addressed my comments with three arguments: First, they pointed out that many of the expanded marker genes are likely not ancient and don't contribute to AB branch length. Second, they found that "as more markers were added (with lower verticality), the inferred AB branch length from the concatenate was reduced." Third, they took marker genes and "concatenated them before and after pruning of unambiguous HGTs", and found that the latter indicated a longer AB branch (but the result is not presented in the manuscript; I apologize if I missed it).These analyses are valuable. However, I am afraid that none of the analyses directly address my comment. To clarify: in order to effectively rebut the concatenation method, one needs to demonstrate that it is the act of concatenation, instead of including genes with lower verticality, that reduces the AB branch length. I understand that it may not be trivial. But since the authors stated that "this (concatenation followed by branch length estimation) is our main critique of the Zhu et al., analyses" (which I agree to certain extent; see my previous comment), robust and compelling analyses are needed to strengthen the point, otherwise it should not be emphasized.

First, we have now added the result mentioned by the reviewer (comparison of concatenated AB branch length before and after pruning HGTs) to the manuscript, as we agree that it may be of value. We now write (in the section on “Finding ancient vertically-evolving marker genes”):

“Further, the AB branch length inferred from a concatenation of the 54 marker genes increased moderately following pruning of recent HGTs, from 1.734 substitutions/site (non-pruned) to 1.945 substitutions/site after manual pruning, consistent with the hypothesis that non-modelled inter-domain HGTs reduce the overall estimate of AB branch length when included in concatenations.”

Second, we were not as clear as we might have been in our previous exchange on this topic. To clarify, we do not mean to rebut concatenation as a method generally: when single gene histories agree, it is a reasonable method for pooling signal from multiple loci. Our point is that the inclusion of genes with discordant histories in the concatenate reduces the overall inferred AB branch length. This reduction occurs because the optimal gene trees for these expanded sets of marker genes feature intermixing of archaea and bacteria, due to hidden paralogy and/or inter-domain gene transfer. They lack a single branch corresponding to the ancestral archaea-bacteria divergence and, when constrained to evolve on a species tree in which the domains are reciprocally monophyletic, support a short inter-domain branch length, in 176 cases as close to zero as possible (Figure 1A). Inclusion of these discordant genes results in a lower concatenated AB branch length in the same way that adding below-average observations to a dataset will result in a lower mean. We have re-read the relevant section of the manuscript carefully and made a small edit to make the sense of a reduction in the overall length estimate clear, writing:

“Overall, these results suggest that genes that recover the reciprocal monophyly of Archaea and Bacteria also evolve more vertically within each domain, and that these vertically-evolving marker genes support a longer AB branch and a greater AB distance. Indeed, Zhu et al., (2019) also recovered a significant positive relationship between gene verticality and relative AB distance (see their Figure 5E). Consistent with these inferences, AB branch lengths estimated using concatenation decreased as increasing numbers of low-verticality markers (that is, markers with higher ∆LL) were added to the concatenate (Figure 1H). These results suggest that inter-domain gene transfers reduce the overall AB branch length when included in a concatenation.”

"## Divergence time estimation"I am glad to see that the authors have significantly expanded the divergence time estimation in response to my and reviewer #3's comments. In particular, Figure 5B is very useful, because it provides a map of inferred evolutionary rate over time and lineages. The authors did a good job in discussing the result in a more neutral and constructive way (DTE is difficult), especially in explaining the outlier in that figure panel (LACA). I agree with the argument in its current form, that the lack of maximum age calibration causes difficulties.However, I would like to note two important observations from Figure 5B: First, the AB branch length revealed by this time-calibrated tree is much shorter than that by the non-time-calibrated trees (e.g., Figures 3 and 6). This pattern is not a good echo of the main point defended in this manuscript (AB branch is long). I wouldn't assert that they contradict, since the branch lengths in the two trees are of different units (geological time vs substitutions per site). However, divergence time is another, and more intuitive definition of "evolutionary distance". In that sense, the short AB distance is not supported by this result.

We agree with the reviewer that AB branch length and divergence time are distinct, and that there is no conflict between the observation of a long AB branch length (measured in substitutions per site) and a (relatively) short geological time between LUCA, LACA, and LBCA. We respectfully disagree with the reviewer’s comment that the AB branch in the time-calibrated tree "is not a good echo of the main point defended in this manuscript (AB branch is long)": indeed, it is precisely this combination of a long AB branch length (as revealed by analyses of vertically-evolving genes, but not the expanded gene set) with fossil evidence that much of that branch length must have accumulated early in Earth’s history (e.g. bacterial fossils at ~3.2Ga demonstrate that the branch from LUCA to LBCA must be <1.295 Gyr) that provides the support for a higher rate (in subs/site/year) during early evolution. This variation in rate implies that a strict clock, or a relaxed clock with a single calibration (as employed by Zhu et al.,), is not sufficient to estimate rates during early evolution.

Second, the result shows that LACA is an outlier in terms of evolutionary rate. This pattern actually supports Petitjean et al., and Zhu et al.,'s hypothesis that there was accelerated evolution during the AB split. Although the authors discussed this result and attributed it to the maximum age calibration, but they did not provide the evolutionary rate estimated without this calibration (LBCA > 3.227Ga). I suspect that in any circumstance, either or both of LACA and LBCA branches would have much higher evolutionary rate than the later diversification of the two domains.

In our view, this summary of Petitjean et al., and Zhu et al.,’s arguments conflates two issues. First, note that Petitjean et al., (2014) and Zhu et al., (2019) argued that ribosomal genes — but, crucially, not all genes — experienced accelerated evolution during the AB split, with the consequence that ribosomal genes may provide an unrepresentative (over-)estimate of the AB branch length. Our analyses do not support that view: for our 54 vertically-evolving genes tracing back to LUCA, we did not observe a statistically significant difference between AB branch lengths for ribosomal and non-ribosomal genes (and indeed the mean ribosomal branch length was moderately shorter, ribosomal marker mean: 1.338, non-ribosomal markers: 2.252).

As to the more general question of whether “there was accelerated evolution during the AB split”, our analyses are compatible with that view. As above, we note that the evidence for a higher evolutionary rate near the root of the tree comes from the combination of (i) a long AB branch length, estimated from ancient vertically-evolving genes and (ii) fossil calibrations implying that the radiation of Bacteria must have been prior to 3.2Ga: the shorter AB branch estimated from the expanded gene set would instead support a lower rate of early evolution. As we note in the manuscript, given the limited availability of calibrations, these estimates are uncertain, and it is also important to consider the role of the underlying model assumptions, which are particularly important when most direct calibration information is shallow in the tree.

Taken together, I think the DTE result presented in this manuscript is quite inline with Zhu et al., despite the different explanations. Therefore I remain reserved on the strength of the authors' argument.

Some disagreement remains but the exchange has been valuable and has helped to improve our manuscript, to clarify where the remaining points of disagreement lie and, perhaps, how they might be resolved in the future. In our view, there is a lot of promise in developing methods that capture processes such as HGT in deep time that will allow more of the data, including families with a range of non-informational functions, to be brought to bear on these questions. More deep fossil calibrations and alternative ways to calibrate the early nodes of the tree would also be of great value in further disentangling evolutionary rate and time at the Archaea-Bacteria divergence.